# Hysteresis and orbital pacing of the early Cenozoic Antarctic ice sheet

Jonas Van Breedam[1], Philippe Huybrechts[1], Michel Crucifix[2]

[1]Earth System Science & Departement Geografie, Vrije Universiteit Brussel, Brussels, Belgium

[2]Earth and Life Institute, Université Catholique de Louvain, Louvain-la-Neuve, Belgium

*Correspondence to*: Jonas Van Breedam (jonas.van.breedam@vub.be)

**Abstract.** The hysteresis behaviour of ice sheets arises because of the different thresholds for growth and decline of a continental-scale ice sheet depending on the initial conditions. In this study, the hysteresis effect of the early Cenozoic Antarctic ice sheet to different bedrock elevations is investigated with an improved ice sheet-climate coupling method that
accurately captures the ice-albedo feedback. It is shown that the hysteresis effect of the early Cenozoic Antarctic ice sheet is about ~180 ppmv or between 3.5˚C and 5˚C, depending only weakly on the bedrock elevation dataset. Excluding isostatic adjustment decreases the hysteresis effect significantly towards ~40 ppmv, because the transition to a glacial state can occur at a warmer level. The rapid transition from a glacial to a deglacial state and oppositely from deglacial to glacial conditions is strongly enhanced by the ice-albedo feedback, in combination with the elevation - surface mass balance feedback. Variations
in the orbital parameters show that extreme values of the orbital parameters are able to exceed the threshold in summer insolation to induce a (de)glaciation. It appears that the long-term eccentricity cycle has a large influence on the ice sheet growth and decline and is able to pace the ice sheet evolution for constant $CO_2$ concentration close to the glaciation threshold.

## 1 Introduction

The concern about nonlinear dynamics in the climate system has recently increased, with the possible collapse of the West-
Antarctic ice sheet (Rosier et al., 2021) and the Greenland ice sheet (Boers and Rypdal, 2021). These so-called tipping points are thresholds beyond which self-reinforcing feedbacks could irreversibly lead to a new state (Lenton et al., 2019; Armstrong McKay et al., 2022). These thresholds might trigger other tipping points and could result in a tipping cascade (Klose et al., 2020). For instance, melting of the Greenland ice sheet could halve the strength of the thermohaline circulation in the North Atlantic when large amounts of meltwater are released from the ice sheet (Van Breedam et al., 2020; Li et al., 2023).

The build-up of the Antarctic ice sheet around the Eocene-Oligocene Transition (EOT) is also a tipping point and occurred at a time that environmental conditions were satisfactory for ice sheet growth, ensuring a series of positive feedbacks, eventually leading to the formation of a large ice sheet. These environmental conditions included the thermal insulation of Antarctica by

either ocean currents (Kennett, 1977), wind patterns (Houben et al., 2019) or both; the decline in carbon dioxide concentrations (Pagani et al., 2011); and tectonic activity that moved the Antarctic continent closer to the South Pole. Geological evidence (Scher et al., 2014; Carter et al., 2017) and modelling work (Van Breedam et al., 2022) also pointed to ephemeral glaciations prior to the EOT. This would imply that thresholds in the climate system were first crossed to initiate and end large-scale glaciations during the late Eocene.

Once environmental conditions are sufficient for ice sheet growth, a small change in climate conditions can trigger a large, nonlinear response. It is generally believed that orbital parameter variations are the cause of large ice sheet changes and have regulated ice sheet growth and decline during almost any geological period in the history of the Earth (Mitchell et al., 2021). These interpretations arise from the identification of periodic events observed in geological archives and are attributed to reflect Milankovitch cycles (De Vleeschouwer et al., 2017). The main influence of changes in the orbital parameters on the climate system is a seasonal and latitudinal redistribution of incoming solar radiation. The different orbital parameters each have a different periodicity. The precession has periods of 19 and 23 kyr, the obliquity of 41 kyr and the eccentricity has periods around 100 kyr and 405 kyr (Laskar et al., 2011). Especially the eccentricity has been thought to play a crucial role on the ice sheet stability (Horton et al., 2012).

Hysteresis is the dependence of a physical system based on its history. The ice sheet-climate system exhibits hysteresis where for a given forcing, either a large ice sheet or a small ice sheet/bare bedrock are stable states. Which one is reached depends on the initial conditions. For instance, when $CO_2$ concentrations are very low, a large continental ice sheet can exist and it will be far from a tipping point (Figure 1). As $CO_2$ concentrations (control parameter) increase, the system gets closer to a tipping point and will eventually make an abrupt change towards a new system state without the ice sheet. The underlying mechanisms to induce the non-linear changes are the ice-albedo feedback and the elevation-surface mass balance feedback.

In previous work, the hysteresis behaviour of ice sheets has been investigated to determine the future thresholds for ice sheet stability of the Greenland ice sheet (Letréguilly et al., 1991; Ridley et al., 2005; Ridley et al., 2010; Robinson et al., 2012; Levermann and Winkelmann, 2016) and the Antarctic ice sheet (Huybrechts, 1994; Garbe et al., 2020). Hysteresis of ice sheet volume variations has been identified during the Quaternary ice ages to explain the 100 kyr cyclicity of prolonged ice sheet build-up and rapid deglaciation (Calov and Ganopolski, 2005; Abe-Ouchi et al., 2013).

Pollard and Deconto (2005) determined the hysteresis of the early Antarctic ice sheet at the Eocene-Oligocene transition using an isostatically rebounded present-day bedrock topography and a matrix method where a limited number of climate model runs were performed based on end members in the forcing. The adopted approach captured the important height-mass balance feedback. However, the ice-albedo feedback may not have been properly taken into account, given the small set of initial ice sheet geometries used in that study. Furthermore, beyond the positive height-mass balance and ice-albedo feedbacks, a negative

feedback from isostasy might arise during the build-up and decline of a continental scale ice sheet (Crucifix et al., 2001). The solid Earth will deform in response to ice mass loading and as a result, the surface elevation increase due to increased accumulation of ice will be delayed. It is proposed that this negative feedback also delays grounding line migration in the marine sectors of Antarctica during the coming centuries (Larour et al., 2019).

In this study, the hysteresis of the early Cenozoic Antarctic ice sheets is tested with an improved methodology, based on an emulator calibrated on 20 predefined ice sheet geometries. This number of different ice sheet geometries allows the climate model to represent the climatic state for a small change in the surface type, being either ice or tundra. This way, the albedo effect from a change in ice sheet area is well-captured. The sensitivity of the threshold behaviour is tested to different recent paleo-bedrock elevation reconstructions. Since isostasy has a potentially significant impact on the glaciation threshold, sensitivity tests are performed to assess the strength of the isostatic adjustment feedback on the ice sheet growth. Additionally, the importance of the orbital parameters on the glaciation and deglaciation thresholds is investigated. Constant forcing simulations are run to explore the influence of the eccentricity, the obliquity and the $CO_2$ values on glaciation and deglaciation thresholds. At the same time, this study also aims at identifying the forcing needed to initiate and end a continental-scale glaciation when the Earth again entered icehouse conditions in the early Cenozoic.

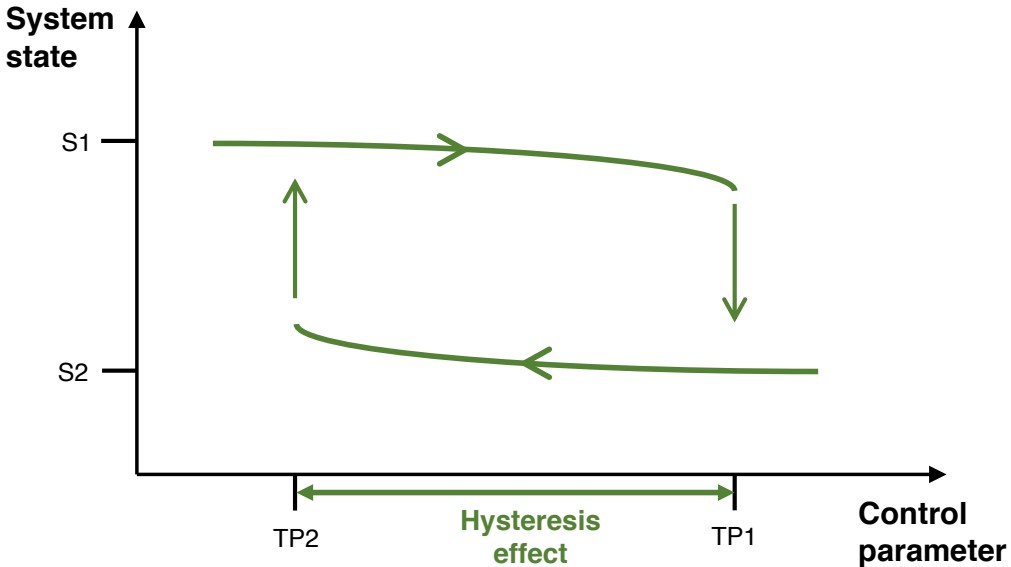

**Figure 1: Typical diagram visualizing the concept of the hysteresis effect. The system state is either a continental-scale ice sheet (S1) or no ice sheet (S2). The control parameter can be the surface temperature or the forcing itself such as the $CO_2$ forcing or the insolation forcing. TP1 and TP2 represent tipping points at which the system can quickly change its state.**

## 2 Model description and experimental set-up

The different models used in this study are the climate model HadSM3 and the ice sheet model AISMPALEO. The models interact with one another through the Gaussian process emulator CLISEMv1.0 (Van Breedam et al., 2021b).

### 2.1 Climate model HadSM3

The coupled atmosphere-slab ocean climate model HadSM3 (Williams et al., 2001) provides the climatic fields. The resolution of the atmospheric component is 2.5˚ for the latitude and 3.75˚ for the longitude. It has a hybrid vertical coordinate with 19 levels in the vertical (Pope et al., 2000). The heat fluxes between the surface and the atmosphere are calculated with the MOSES-1 scheme (Cox et al., 1999). The slab ocean model equilibrates with the atmosphere in a 50 m thick layer where heat is exchanged with the atmosphere. In slab ocean-atmosphere models, an anomalous heat convergence is added to the ocean to mimic the real influence of oceanic circulation below the slab layer. Monthly variable, zonal mean sea surface temperatures (SSTs), representative for the late Eocene (Evans et al., 2018), are prescribed to calibrate the anomalous heat convergence. These heat fluxes, representative for the horizontal heat transport and seasonal deep-water exchange, are then used to simulate realistic sea surface temperatures under various climate conditions. Sea ice is simulated during the winter months and is mostly constrained to the Weddell Sea region, but disappears again during the austral summer in most simulations. The palaeogeographic reconstruction is based on the method presented in Baatsen et al. (2016) using the van Hinsbergen et al. (2015) plate rotational model. The bedrock topography used in the climate model is the Wilson maximum bedrock topography (Wilson et al., 2012) and is representative for the Eocene-Oligocene transition (EOT) at 34 Ma (Figure 2). The minimum and maximum bedrock topographies are applied as a boundary condition in the ice sheet model at a 40 km resolution. In order to grasp the entire uncertainty, each ice sheet model grid cell takes on the lowest and highest value for respectively the minimum and maximum bedrock topography from the original higher resolution Wilson et al. (2012) dataset within each ice sheet grid cell.

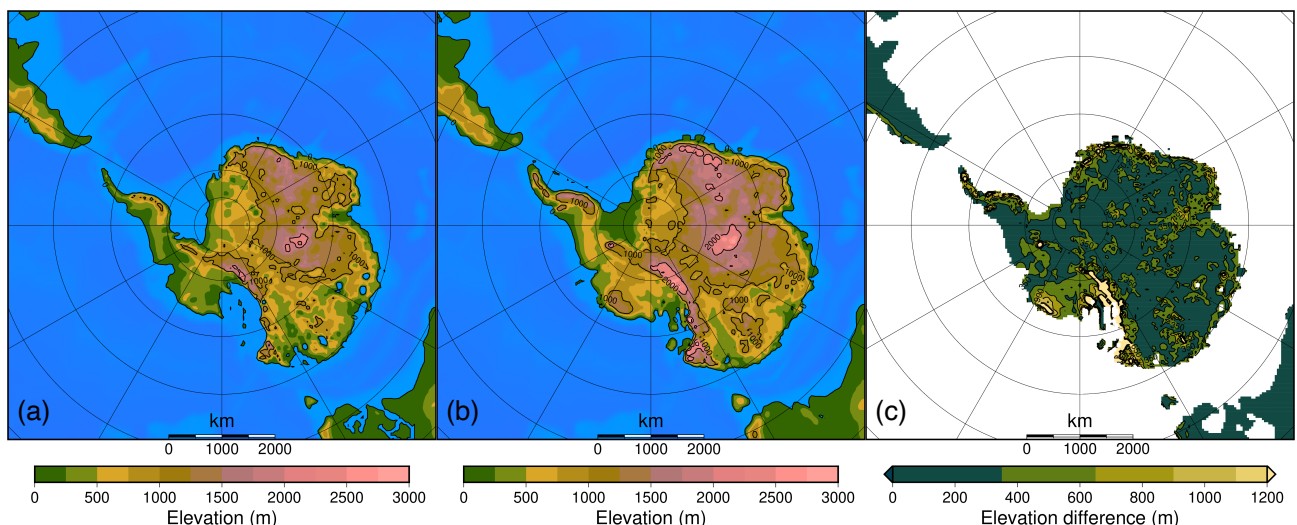

**Figure 2: (a) Minimum and (b) maximum bedrock elevation reconstructions from Wilson et al. (2012). (c) The elevation difference between the maximum and minimum bedrock reconstructions.**

## 2.2 Emulator CLISEMv1.0

The emulator CLISEMv1.0 (Van Breedam et al., 2021b) is used to force the ice sheet model during the multi-million-year simulations. The emulator is calibrated on 100 preliminary climate model runs from HadSM3, which is forced with a unique combination of the orbital parameter ε (obliquity) and the parameter combinations $esin\varpi$ and $ecos\varpi$, where $e$ is the eccentricity and $\varpi$ is the longitude of the perihelion. The 100 climate model runs also sample 20 different ice sheet geometries, ranging from a very small ice cap in the Gamburtsev mountains up to a fully glaciated Antarctic ice sheet. The geometries have been chosen such that their ice volumes have a good spread between the smallest and the largest ice sheet geometries. The use of 20 different ice sheet geometries is equivalent to grasping the surface type differences at the resolution of the climate model and therefore, the albedo changes can be fully captured. The albedo varies between the discrete values of 0.8 for ice/snow and 0.2 for tundra. The $CO_2$ forcing across the runs ranges from 550 ppmv to 1150 ppmv. After a calibration and validation process of the 100 preliminary climate model runs (see for details Van Breedam et al., 2021a), the emulator is able to provide the climatic forcing (temperature and precipitation fields) necessary to drive the mass balance of AISMPALEO for any combination of the orbital parameters, the $CO_2$ level and the ice sheet volume (Eq. 1). The orbital parameter combinations (Laskar et al., 2011) and the $CO_2$ concentration are prescribed, while the ice sheet volume ($V_{ice}$) is calculated within the ice sheet model. The emulator set-up is the same as EMULATOR_20 calibrated on ice volume as presented in Van Breedam et al (2021a) which had been applied to the late Eocene to investigate the potential for late Eocene glaciations (Van Breedam et al., 2022). The climatic fields are updated every 500 years in the ice sheet model, a time-step that is sufficient to capture the feedbacks between the ice sheet and the external forcing (Herrington and Poulsen, 2011).

$$\begin{Bmatrix} T \\ precip \end{Bmatrix} = f(esin\widetilde{\omega}, ecos\widetilde{\omega}, \varepsilon, CO_2, V_{ice}) \quad (1)$$

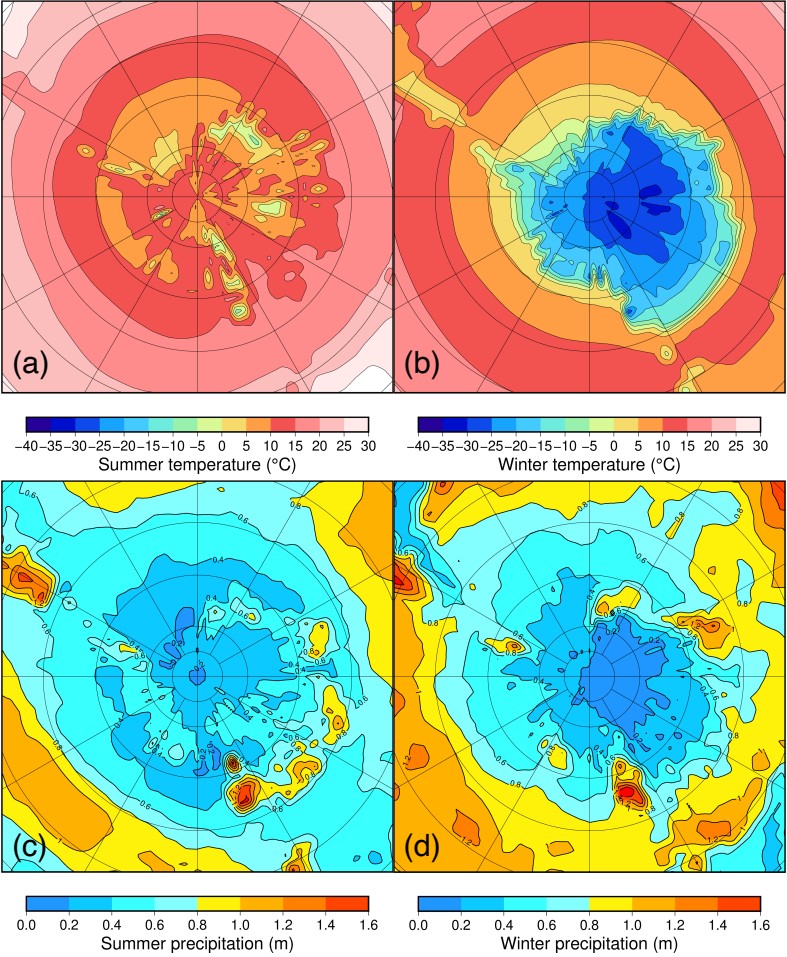

**Figure 3: Antarctic climatologies for a 2x pre-industrial CO$_2$ forcing (560 ppmv). (a) Summer (DJF) temperature. (b) Winter (JJA) temperature. (c) Summer (DJF) precipitation. (d) Winter (JJA) precipitation.**

135

The simulated Antarctic climate is strongly dependent on the forcing. In Fig. 3, seasonal temperature and precipitation patterns are shown for a nearly ice-free Antarctica having a few small ice caps on the highest elevations only. Here, the CO$_2$ forcing is representative of a 2x pre-industrial CO$_2$ forcing (560 ppmv), the obliquity is close to the present-day obliquity with 23.4˚, the eccentricity is very small (0.007) resulting in an almost circular orbit and therefore, the effect of the precession is negligible.

140 The precipitation is largest along the Antarctic continental margin and decreases inland. There is a lower precipitation rate during the austral winter (June-July-August or JJA) than during the austral summer (December-January-February or DJF)

above the Antarctic continent. In winter, most precipitation falls as snow, while snowfall is limited to the highest elevated regions such as the Gamburtsev Mountains and Dronning Maud Land in summer. The seasonality is extremely large on the Antarctic continent with a winter-summer difference in mean daily temperatures of 30-35 ˚C along the continental margin up to 50-55˚C in the interior of the continent close to the South Pole. This is in line with the study from Baatsen et al. (2020) using CESM 1.0.5, who identified a large seasonality of 35-45˚C on the Antarctic continent and a monsoonal type of climate with larger summer precipitation.

## 2.3 Antarctic ice sheet model AISMPALEO

The three-dimensional thermomechanical ice sheet model AISMPALEO is used at a resolution of 40 km. The model has 30 levels in the vertical, with a closer spacing towards the bedrock where most of the shearing occurs. Grounded ice flow is calculated using the Shallow Ice Approximation (SIA), while ice shelf flow is derived from the Shallow Shelf Approximation (SSA) in case the ice sheet reaches the ocean. Grounded ice flow is a result from internal deformation and basal sliding at locations where the pressure melting point is reached (Huybrechts and de Wolde, 1999). The basal sliding velocity in AISMPALEO follows a Weertman relation and is proportional to the third power of the basal shear stress and inversely proportional to the height above buoyancy. The basal sliding coefficient is a constant multiplication factor for the basal sliding and equals $1.8 \times 10^{-10}$ $N^{-3}yr^{-1}m^8$. The sensitivity of the ice sheet model to ice sheet parameter uncertainties are not explored. An enhancement factor of 1.8 is used for grounded ice. This is similar to the value used to model the present-day Antarctic ice sheet with AISMPALEO. A constant geothermal heat flux of 50 mW $m^{-2}$ has been applied over the entire model domain.

The grounding line is a one grid cell wide transition zone between the grounded and floating ice where all the stress components contribute in the effective stress in the flow law. Ice shelves develop when the grounded ice reaches the coast and the influx of snow from the atmosphere and ice from upstream exceeds the sum of surface ablation and basal melting. Although the slab ocean model exchanges heat with the atmosphere and records changes in the sea surface temperature, we do not use this information to calculate basal melt rates. Instead, we prescribe a constant basal melt rate of 10 m $yr^{-1}$ over the entire domain. This is a strong simplification (and perhaps it is even too low in some locations). It allows for small ice shelves to develop once the ice sheet reaches the coast. Nevertheless, even for present-day simulations large uncertainties exist in how changes in ocean temperature and salinity affect melt rates below the ice shelves (Burgard et al., 2022). There is no special treatment of calving fronts and iceberg calving occurs when the ice shelf front is thinner than 150 m. .

The surface mass balance is computed using the efficient Positive Degree Day (PDD) model (Janssens and Huybrechts, 2000). The PDD model uses the yearly sum of the mean daily temperatures above 0˚C to determine the melt potential. In practice monthly time steps are sufficient to calculate the total amount of PDD's. Random weather fluctuations and the daily cycle in temperature are taken into account by including a standard deviation of the mean daily temperature of 4.2˚C. Different Degree

Day Factors (DDF) are used to melt snow and ice, respectively 0.003 m ice equivalent (i.e.) $°C^{-1}$ $day^{-1}$ and 0.008 m i.e. $°C^{-1}$

$day^{-1}$. The melt rate is computed by multiplying the DDFs of snow and ice with the sum of the PDDs. The difference between snow and rain is calculated based on a surface temperature threshold of 1˚C. When ice is melting, part of the meltwater can be retained or refreeze in the snowpack to form superimposed ice.

The model includes a component to calculate isostatic adjustment due to ice mass addition and removal. The isostatic model

consists of an elastic lithosphere with a flexural rigidity D of $10^{25}$ Nm (which is a measure of the strength of the lithosphere) on top of a viscous asthenosphere, to allow the crust to deform far beyond the local ice loading (Huybrechts, 2002). The vertical deflection of the lithosphere w is given by a fourth order differential equation (Eq. 2) Here, q is the ice load, $\rho_m$ is the mantle density (3300 kg $m^{-3}$) and g the gravitational acceleration. This equation is solved using a Kelvin function of zero order (kei). The viscous asthenosphere responds to the ice load with a relaxation time τ of 3000 years (Le Meur and Huybrechts, 1996).

$$D\nabla^4 = q - \rho_m g\,w \quad (2)$$

## 2.4 Experimental design

All simulations span the time period between 40 Ma and 30 Ma, for which the orbital forcing for the different simulations is provided. Because the initial bedrock topography has a large influence on the ice sheet inception and also on the size of the

continental scale ice sheet, simulations are performed for the minimum and maximum bedrock topography estimates from Wilson et al. (2012) in order to test the hysteresis and threshold behaviour for two extremes (Figure 2 and S1). Sea-level changes (sea-level fall when the ice sheet is growing) are not included as an additional forcing because the number of grid points that become land-based when sea-level would fall by -70 m are very limited. The simulations that start from a bare bedrock are forced with an atmospheric $CO_2$ concentration linearly decreasing from 1150 ppmv to 550 ppmv. The resulting

continental scale ice sheet extent is then used for the simulations starting from a glaciated continent and the $CO_2$ concentration is linearly increased from 550 ppmv to 1150 ppmv (Table 1). The lower and upper bounds of the $CO_2$ concentration are chosen to include a forcing when either the ice sheet grows to a continental scale for any astronomical forcing (550 ppmv) and a forcing that is too high to allow for any ice sheet to exist (1150 ppmv) in our model set-up. Additional sensitivity experiments are performed in which isostatic adjustment is not considered.


**Table 1: Standard set of experiments with the Wilson minimum and Wilson maximum bedrock topography as boundary conditions and variable orbital forcing.**

| Bedrock topography | Ice level (at start) | $CO_2$ (ppmv) | Eccentricity | Obliquity | Isostatic adjustment | Experiment duration |
|---|---|---|---|---|---|---|
| Wilson minimum | No ice | 1150 to 550 (linear) | variable | variable | Yes | 10 Myr |

| Wilson minimum | Ice | 550 to 1150 (linear) | variable | variable | Yes | 10 Myr |
| Wilson maximum | No ice | 1150 to 550 (linear) | variable | variable | Yes | 10 Myr |
| Wilson maximum | Ice | 550 to 1150 (linear) | variable | variable | Yes | 10 Myr |
| Wilson maximum | No ice | 1150 to 550 (linear) | variable | variable | No | 10 Myr |
| Wilson maximum | Ice | 550 to 1150 (linear) | variable | variable | No | 10 Myr |

An additional set of runs (Table 2) explores the variation in orbital forcings to investigate the influence of the insolation
thresholds for ice sheet growth and decline in detail. In these runs, different values for the individual orbital parameters or the
insolation are the control parameters explored (see Figure 1 and Table 2) and the $CO_2$ concentrations are kept constant. The
orbital parameter solution La2010 from Laskar et al. (2011) is used (Figure 4). The orbital parameters contain periods ranging
from about 20 kyr for the precession up to 405 kyr for the long-term cycle in the eccentricity. On a multi-million-year timescale,
the eccentricity also exhibits variations with a longer period. Spectral analyses reveal that this cycle has a main periodicity of
2.4 Ma (Laskar et al., 2004). The duration of the experiments where the influence of the eccentricity is investigated is 2.4 Myr
long starting at 34.2 Ma up to 31.8 Ma, to capture the extrema when the 100 kyr, 405 kyr and 2.4 Ma cycles reach a maximum,
separately and combined. In the latter experiments, the $CO_2$ forcing is explored in a narrow window of 80 ppmv at an interval
of 10 ppmv. The other experiments where the eccentricity is constant and the obliquity is variable or the obliquity is constant
and the eccentricity is variable are sampled in a larger $CO_2$ window of 450 ppmv at an interval of 50 ppmv (Table 2). These
experiments have a duration of 200 kyr because they equilibrate faster with the forcing (there is a limited influence of changes
in the orbital forcing).

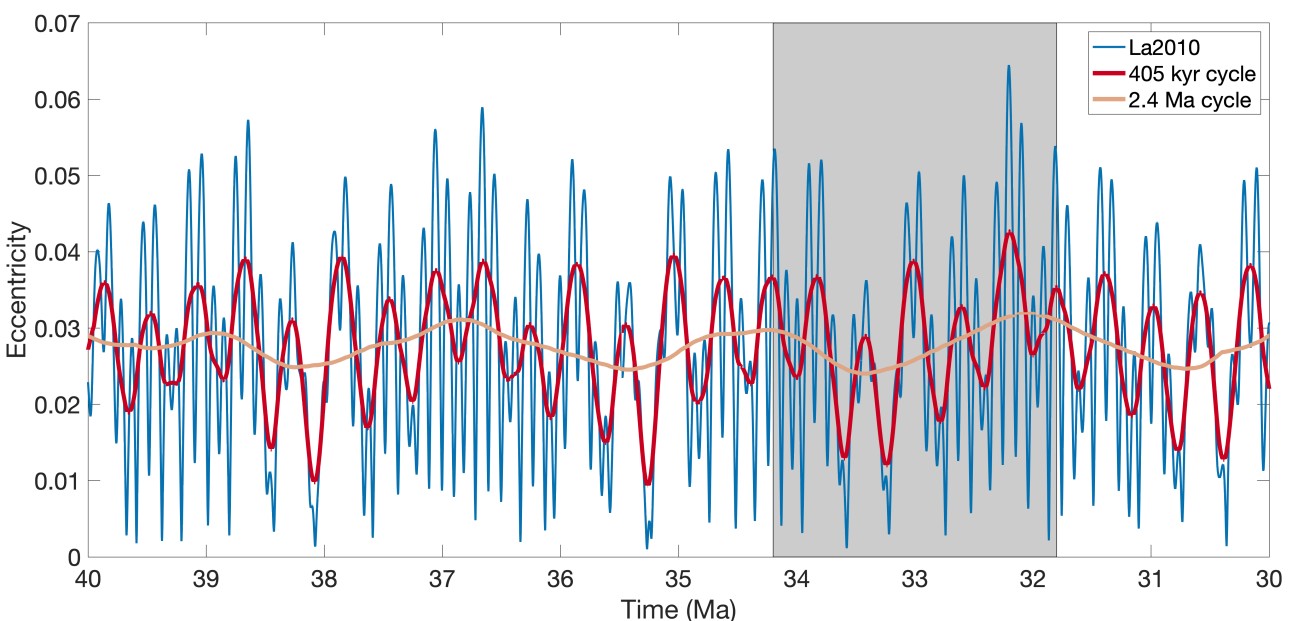

**Figure 4: Earth's orbital eccentricity between 40 Ma and 30 Ma using the La2010 orbital solution (Laskar et al., 2011). The ~100 kyr cycle (blue) is clearly visible. The 405 kyr cycle (red) and 2.4 Ma cycle are visualized by applying a running mean on the eccentricity from the La2010 solution. The period between 34.2 Ma and 31.8 Ma (grey shaded region) indicates one 2.4 Myr period in eccentricity at around the EOT.**

**Table 2: Experiment overview for the runs investigating the influence of the orbital parameters for fixed $CO_2$ concentration levels.**

| Experiment | Bedrock topography | Ice level (at start) | $CO_2$ (ppmv) | Eccentricity | Obliquity | Isostatic adjustment | Experiment duration |
|---|---|---|---|---|---|---|---|
| (a) | Wilson maximum | Ice | 980, 990, 1000, 1010, 1020, 1030, 1040, 1050, 1060 | Variable | Variable | Yes | 2.4 Myr |
| (b) | Wilson maximum | No ice | 810, 820, 830, 840, 850, 860, 870, 880, 890 | Variable | Variable | Yes | 2.4 Myr |
| (c) | Wilson maximum | Ice | 650, 700, 750, 800, 850, 900, 950, 1000, 1050, 1100, 1150 | 0.01, 0.02, 0.03, 0.04, 0.05, 0.06 | variable | Yes | 200 kyr |
| (d) | Wilson maximum | No ice | 650, 700, 750, 800, 850, 900, 950, 1000, 1050, 1100, 1150 | 0.01, 0.02, 0.03, 0.04, 0.05, 0.06 | variable | Yes | 200 kyr |
| (e) | Wilson maximum | Ice | 600, 650, 700, 750, 800, 850, 900, 950, 1000, 1050, 1100, 1150 | variable | 22.5°, 23°, 23.5°, 24°, 24.5° | Yes | 200 kyr |
| (f) | Wilson maximum | No ice | 600, 650, 700, 750, 800, 850, 900, 950, 1000, 1050, 1100, 1150 | variable | 22.5°, 23°, 23.5°, 24°, 24.5° | Yes | 200 kyr |

## 3 Ice sheet hysteresis

The thresholds for glaciation and deglaciation depend on the bedrock topography, because of its influence on surface elevation, and hence, on surface temperature. In section 3.1 we first test the sensitivity of the Antarctic ice sheet hysteresis to the initial bedrock topography dataset. Additionally the influence of the isostatic adjustment during the build-up of an ice sheet is quantified in section 3.2.

### 3.1 Sensitivity to the bedrock topography

Antarctic continental scale glaciation is strongly sensitive to the prescribed bedrock topography dataset. Fig. 5 shows that the $CO_2$ concentration threshold to initiate Antarctic glaciation is higher when the bedrock topography is higher with a value of 870 ppmv for the maximum bedrock topography reconstruction and a value of 650 ppmv for the minimum bedrock topography reconstruction. This is a logical consequence of the temperature decrease with elevation. The snowline will intersect the topography more easily for higher $CO_2$ values when the initial bedrock topography is higher. The same is true for the ice sheet's decline. The higher the initial bedrock topography, the larger the final extent and elevation of the ice sheet. The mean annual surface temperature (MAT_sur), defined here as the mean for all the land-based and ice-covered grid points, is lower for a geometry with a higher fraction of ice sheet grid points because of the resulting higher mean elevation. Hence, to melt a continental scale ice sheet with a larger area and a higher surface elevation, the $CO_2$ concentration must be higher. The difference in the $CO_2$ threshold or the hysteresis effect between glaciation and deglaciation does not depend much on the bedrock topography. For the Wilson maximum topography, the $CO_2$ threshold difference between glaciation and deglaciation is about 180 ppmv, while the threshold is 170 ppmv for the Wilson minimum topography.

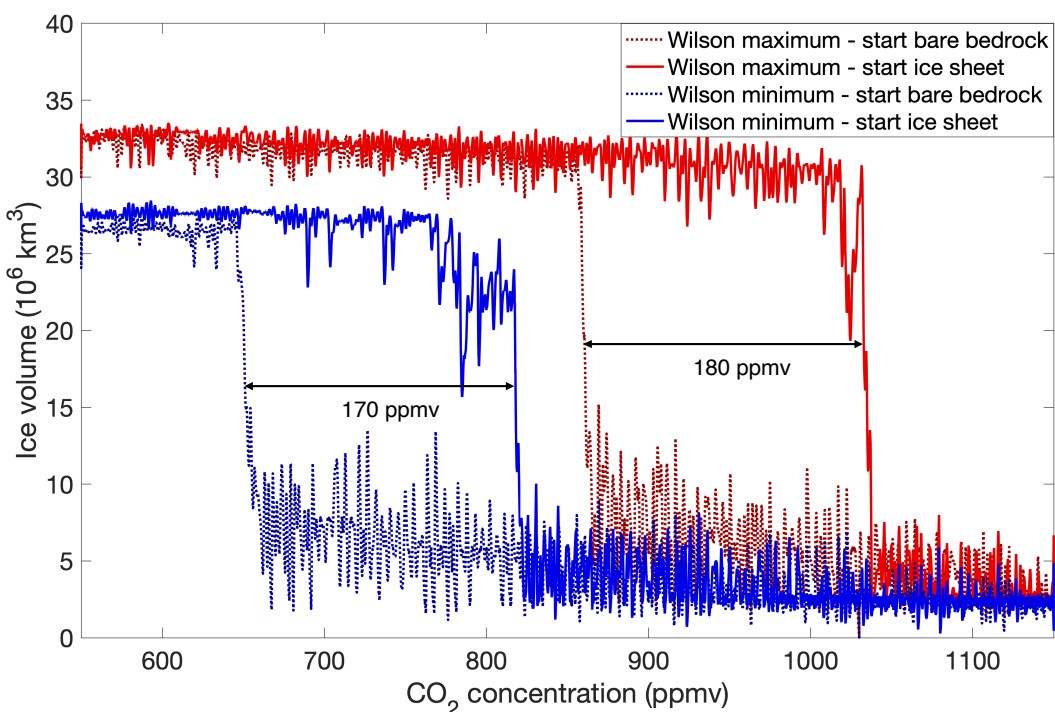

**Figure 5: Hysteresis behaviour of ice sheet growth and decline for linearly varying $CO_2$ concentrations between 550 ppmv and 1150 ppmv for the Wilson maximum bedrock topography (red) and the Wilson minimum bedrock topography (blue) reconstructions. The dotted lines represent simulations that start from a bare bedrock topography. The solid lines represent simulations that start from a continental-scale ice sheet.**

The difference between the glaciation and deglaciation thresholds can also be expressed in terms of a temperature change. Grounding line retreat for marine-based ice sheets can be initiated by subsurface melting of ice shelves, but melting of a continental scale ice sheet is mostly governed by changes in the surface mass balance. Hence, the surface temperature is key in determining the threshold at which the ice sheet starts to grow or when the deglaciation occurs. MAT_sur defined for individual grid points ranges from -5°C at the ice sheet margin to between -45°C and -50°C in East Antarctica for respectively
the Wilson minimum and Wilson maximum bedrock topography. The colder temperatures for the Wilson maximum topography are mainly due to the differences in surface elevation (Figure 6). These temperatures are reached for a $CO_2$ forcing of 550 ppmv and are still about 10-20°C higher than the temperatures over the present-day Antarctic ice sheet.

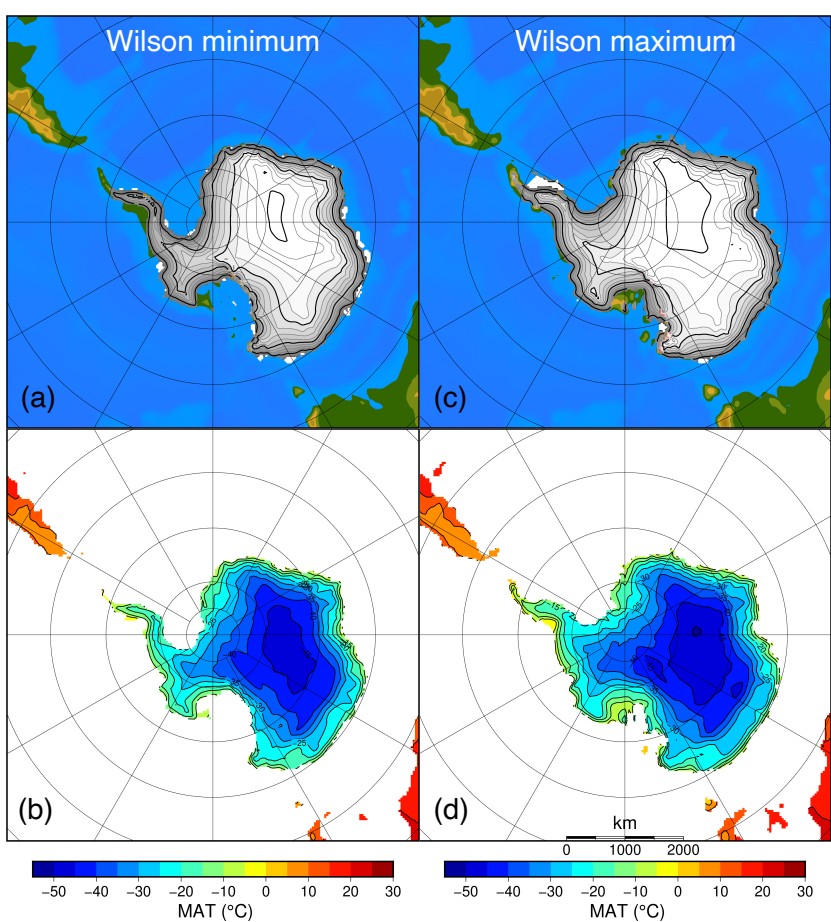

**Figure 6: Continental-scale ice sheet geometry for the (a) Wilson minimum bedrock topography and (b) the corresponding spatially varying mean annual surface temperature and for (c) the Wilson maximum bedrock topography and (d) the corresponding spatially varying mean annual surface temperature. Thin contour intervals are given every 250 m, while thick contour intervals are given each 1000 m for the ice sheet surface elevation.**


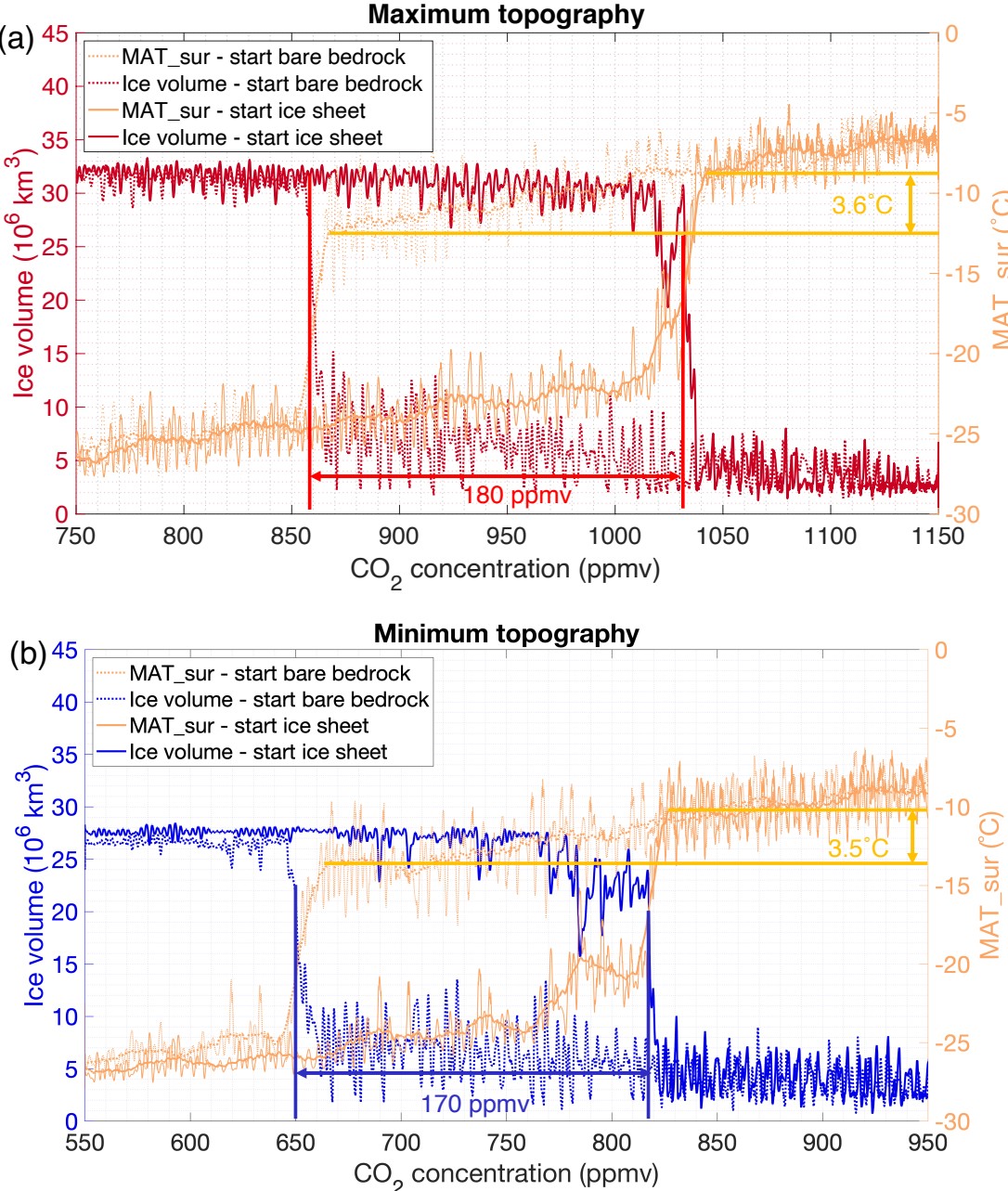

**Figure 7: Mean annual temperature change (MAT_sur) and ice volume evolution using (a) the Wilson maximum bedrock topography and (b) the Wilson minimum bedrock topography. The simulation starts either from a bare bedrock and a CO₂ forcing of 1150 ppmv that linearly decreases towards 550 ppmv or from a continental scale ice sheet and a CO₂ forcing of 550 ppmv that linearly increases towards 1150 ppmv.**

The MAT_sur for a deglaciated Antarctic continent at 1150 ppmv is about -7°C during the late Eocene for the Wilson maximum bedrock topography (Figure 7a) and slightly higher with -6°C for the Wilson minimum bedrock topography (Figure 7b). The annual march in temperatures is about 30°C with a mean January temperature of 13°C and a mean July temperature of -17°C (not shown). This indicates that in winter time snow accumulates, while it melts again in summer. The MAT_sur needs to be lower to cause a full glaciation, by ~6°C and ~8°C respectively for the Wilson maximum and minimum bedrock topographies. Temperatures range from -26°C to -29°C for the Wilson minimum and the Wilson maximum bedrock topography at a $CO_2$ concentration of 550 ppmv for a fully glaciated continent. To make the transition back to a deglaciated continent, the MAT_sur needs to rise above -20°C (Figure 7).

For the Wilson minimum bedrock topography, the MAT_sur above the Antarctic continent is -14°C when the entire continent is glaciated starting from a bare bedrock. Starting from a fully glaciated continent, the ice sheet deglaciates when the MAT_sur reaches -10°C. For the Wilson maximum bedrock topography, these temperatures are very similar with a MAT_sur of -13°C when the entire continent is glaciated starting from a bare bedrock and deglaciation starts when the MAT_sur reaches -9°C. This again indicates that the hysteresis effect is quasi-independent of the particular bedrock elevation dataset. The hysteresis effect, expressed in terms of the MAT_sur threshold between glaciation and deglaciation, is about 4°C.

### 3.2 Influence of isostasy

It is thought that isostatic adjustment provides a negative feedback to ice sheet growth. During the build-up of the ice sheet, the ice mass interacts with the lithosphere and deforms the underlying bedrock. Hence, the ice sheet elevation will not rise as fast and the increase in accumulation area (when the ice sheet's height exceeds the snowline) will be delayed. The magnitude of this feedback is determined by performing experiments in which isostatic adjustment is not considered.

The hysteresis effect in terms of the $CO_2$ forcing is much less pronounced when isostasy is neglected. The ice sheet decline occurs for a similar threshold of about 1030 ppmv when isostatic adjustment is not taken into account (Figure 8). This is mainly a consequence of the ice-albedo feedback that is similarly strong in both the experiment with isostasy and without isostasy because the area of the continental scale ice sheet is the same regardless of the inclusion of isostatic adjustment. The maximum height of the equilibrated continental-scale ice sheet is about 160 m higher without considering isostasy, while the ice thickness differences exceed 1000 m (Figure S2). The small difference in surface elevation barely has an impact on the $CO_2$ threshold to induce the deglaciation.

Without considering isostasy, the threshold towards full glaciation occurs at a $CO_2$ concentration of 990 ppmv (compared to 850 ppmv for the simulation including isostasy) due to the more rapid increase of the surface elevation when snow accumulates. Aside from the higher surface elevation (up to 800 m higher) prior to the glaciation threshold, the ice sheet area is about twice

305 the size when excluding isostasy. In this case, the hysteresis effect with respect to the $CO_2$ forcing is only 40 ppmv. The MAT_sur threshold to glaciation is again about -13˚C, exactly the same temperature threshold as the experiment including isostasy. As already stated, this is a consequence of the increase of surface elevation when snow and ice accumulate. The deglaciated continent has a MAT_sur of -6˚C, because of the smaller ice sheet extent and generally lower surface elevation (Figure 9).

310

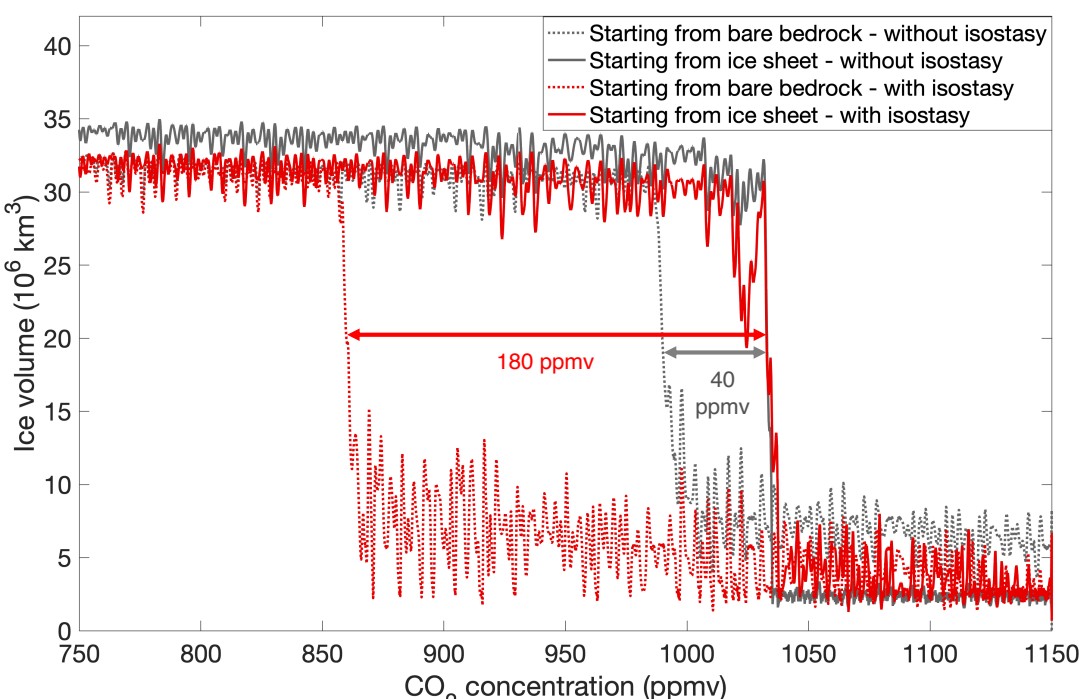

**Figure 8: Ice sheet growth and decline for linearly decreasing/increasing $CO_2$ concentrations from 750 ppmv to 1150 ppmv for the Wilson maximum bedrock topography reconstruction. The grey lines represent the ice sheet evolution for the simulations excluding isostasy and the red lines represent the ice sheet evolution with isostasy.**

315

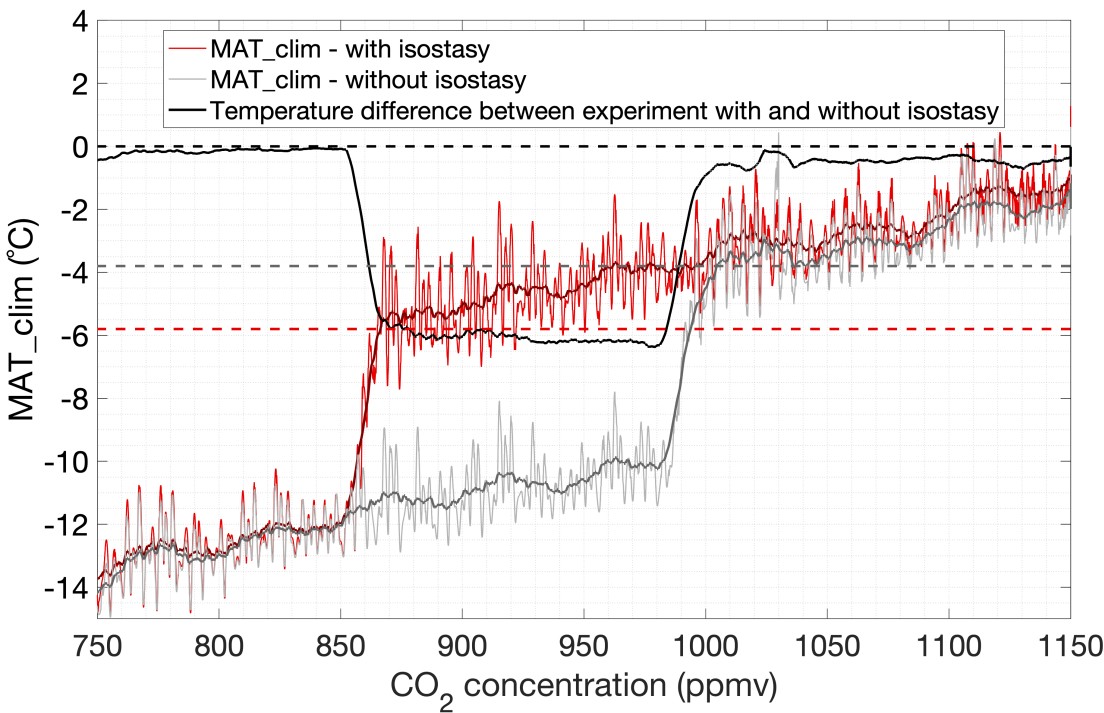

**Figure 9: Elevation corrected mean annual temperature (MAT_clim) above the Antarctic continent. The simulations start from a bare bedrock and either include isostasy (red) or exclude isostasy (grey). The black dashed horizontal line indicates no temperature difference between the experiment including or excluding isostasy. The red and grey horizontal dashed lines indicate the MAT thresholds at which the Antarctic ice sheet grows to a continental scale.**

The temperature threshold to glaciation and deglaciation is very similar for the experiments that include isostatic adjustment and those that exclude it. To disentangle the influence of surface elevation, the MAT_sur is now corrected for the surface elevation change by applying a constant lapse rate correction in order to calculate the temperature at sea-level, which we equate to the climatological mean annual temperature (MAT_clim). As it occurs, the initial temperature difference of about 0.5˚C between both experiments with $CO_2$ concentrations between 1000 and 1150 ppmv is due to the larger ice sheet area in the experiment that excludes isostasy. After the transition to a continental scale ice sheet for a $CO_2$ concentration below 850 ppmv, this difference between both simulations is negligible because the area of the continental scale ice sheet is ultimately bounded by the size of the Antarctic continent. The larger the ice sheet area, the more incoming solar radiation will be reflected and the lower the surface temperature will be. The difference in the elevation corrected MAT_clim threshold to full glaciation is ~2˚C (Figure 9).

## 4 Threshold dependency on orbital forcing

Ice sheet hysteresis occurs when a threshold in the forcing is crossed that leads to a self-amplifying positive feedback loop. On palaeoclimatic timescales on the order of 100 kyr and longer, this threshold is determined by changes in the carbon cycle (or changes in the atmospheric $CO_2$ concentration) and variations in the orbital parameters. First, the influence of the eccentricity is evaluated because the eccentricity modulates the magnitude of the precession and hence determines the magnitude of a threshold (Section 4.1).Then, simulations are run to a steady-state for a constant forcing for different orbital parameter combinations (Section 4.2).

### 4.1 Influence of the eccentricity

There is a different $CO_2$ threshold to continental scale glaciation for each tested bedrock topography. However, in a narrow range of $CO_2$ concentrations, the glaciation threshold is also paced by the orbital parameters. Fig. 10 shows the eccentricity thresholds to continental-scale glaciation for a constant $CO_2$ forcing between 810 ppmv and 890 ppmv and the eccentricity thresholds for complete deglaciation for a $CO_2$ forcing between 980 and 1060 ppmv. All simulations start with the same initial conditions and the ice sheet volume responds to the orbital forcing (experiments (a) and (b) in Table 2). Throughout the 2.4 Myr long runs, the ice sheet geometry has changed at each eccentricity extremum (~100 kyr, ~400 kyr, ~2.4 Myr) because the ice sheet size reacts periodically to the precession (~20 kyr). The time axis indicates the time needed since the start of the simulations to reach the eccentricity threshold to glaciation or deglaciation. For atmospheric $CO_2$ values below 810 ppmv, the ice sheet always reaches a continental scale when the eccentricity declines below 0.04. For $CO_2$ values exceeding 890 ppmv, the ice sheet cannot grow to a continental scale, not even during a prolonged insolation minimum. In this narrow range of $CO_2$ concentrations, the eccentricity has a large influence on the ice sheet initiation and stability. Starting from a bare bedrock, the ice sheet stops growing and either declines or stays with constant volume whenever the eccentricity exceeds a value of 0.05 and then grows again when eccentricity declines. The ice sheet can initially grow and vary with the precession for up to four cycles before the onset to full glaciation sets in, depending on the value for the eccentricity and the $CO_2$ forcing (Figure S3). This shows that it is not only the magnitude of the eccentricity that determines the timing of ice sheet growth, but also the initial size of the ice sheet that is determined by the ice sheet history. Even though the $CO_2$ forcing is constant, the ice sheet grows to a fully glaciated state consecutively for a $CO_2$ forcing of 850 ppmv, 860 ppmv, 870 ppmv, 880 ppmv and 890 ppmv. The higher the $CO_2$ concentration, the more the ice sheet needs to grow before the threshold to full glaciation is reached.

The extreme eccentricity value of 0.064 that occurs after 2 Myr is not sufficient to melt the ice sheet once it has grown to a continental scale for a $CO_2$ forcing between 810 and 890 ppmv. This shows again the hysteresis effect of the Antarctic ice sheet: an eccentricity below 0.032 is enough to initiate continental-scale ice sheet growth for a $CO_2$ forcing up to 890 ppmv, while an eccentricity of >0.06 is not sufficient to make the ice sheet melt entirely after this. The extreme eccentricity of 0.064 influences the ice sheet volume for all simulations forced with a constant $CO_2$ concentration between 800 and 890 ppmv, but

the peak insolation forced by the precessional cycle is too short-lived to melt the entire ice sheet (Figure S3). Such extreme values of the eccentricity exceeding 0.06 only occur when the 100 kyr cycle, the 405 kyr cycle and the 2.4 Ma cycle reach a maximum. The absolute maximum in the eccentricity of 0.064 at 32.2 Ma and a minimum in the 2.4 Ma cycle around 33.4 Ma are captured in this interval, important for determining the respective thresholds to ice sheet decline and ice sheet growth.

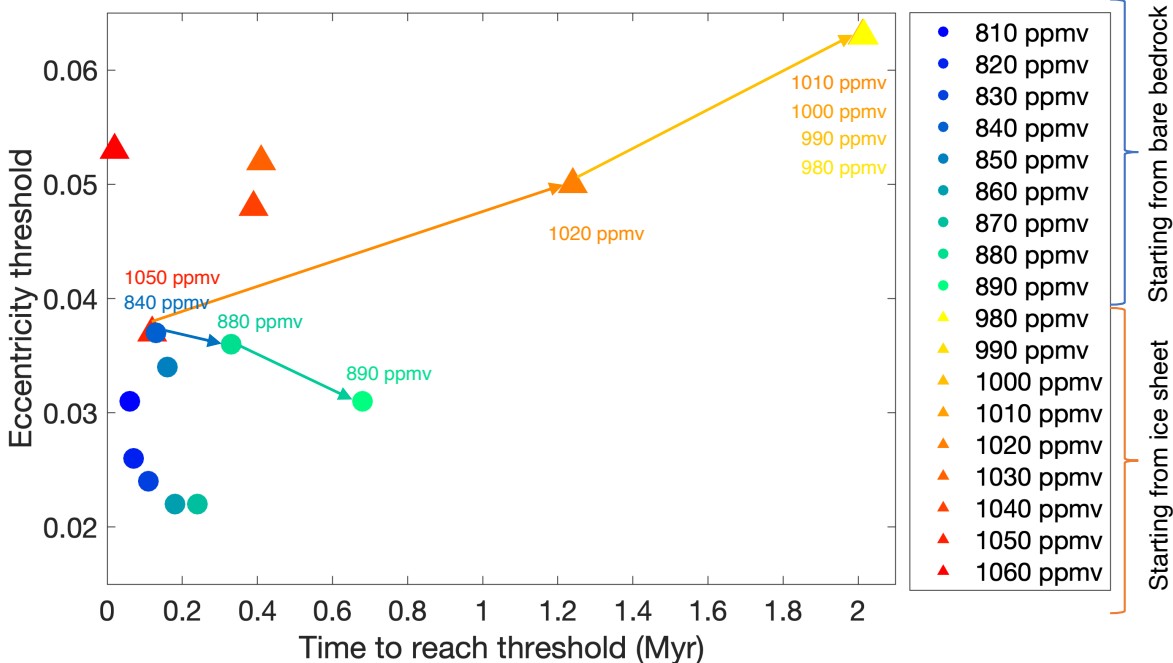


**Figure 10: Eccentricity thresholds for ice sheet initiation starting from a bare bedrock (blue to green dots) and eccentricity thresholds for ice sheet decline (red to yellow triangles) for a range of different $CO_2$ values. The time on the x-axis indicates the duration before the continental-scale ice sheet growth or the ice sheet decline initiates. The experiment duration has an influence on the initial size of the ice sheet before another threshold is reached. The blue**
**and green arrows indicate the decrease in eccentricity threshold for increasing $CO_2$ values to initiate glaciations. The orange and yellow arrows indicate the increase in the eccentricity threshold for decreasing $CO_2$ values to end glaciations. The thresholds are determined using the Wilson maximum bedrock topography.**

Starting from a fully glaciated continent, the $CO_2$ threshold to complete deglaciation has a range between 980 and 1060 ppmv
(Figure 10). An eccentricity of >0.05 initiates ice sheet decline in a $CO_2$ concentration range between 1020 and 1060 ppmv. Either a higher value for the eccentricity or a longer duration since the start of the experiments is needed to disintegrate the Antarctic ice sheet for lower $CO_2$ values. The experiment duration has an influence on the initial size of the ice sheet. For instance, the eccentricity maximum of 0.051 at the start of the experiment was not high enough to make the Antarctic ice sheet disappear at a $CO_2$ forcing of 1050 ppmv. However, the ice sheet declines during the next eccentricity maximum exceeding

0.038 for a $CO_2$ forcing of 1050 ppmv, while it can regrow for a $CO_2$ forcing of 1040 ppmv. This indicates that the $CO_2$ and orbital parameter thresholds exhibit state-dependency; the thresholds to glaciation or deglaciation differ depending on the initial size of the ice sheet. The extreme eccentricity of 0.064 also induces the deglaciation at $CO_2$ concentrations between 980 ppmv and 1010 ppmv.

## 4.2 Constant forcing runs for different orbital parameter combinations

So far, the hysteresis effect is investigated with a time-dependent forcing. To further separate the effects of the orbital parameters and the $CO_2$ thresholds on glaciation and deglaciation, a number of simulations are run to a steady-state for a constant forcing. Three different summer insolation (daily mean for DJF at 65˚S) values are selected to perform the steady-state runs: a relative low summer insolation of 427 $Wm^{-2}$ (further referred to as a 'cold' orbital configuration), a relative high summer insolation of 467 $Wm^{-2}$ (further referred to as a 'warm' orbital configuration) and an insolation value in between these two extremes of 440 $Wm^{-2}$ (Figure 11). These insolation values can be achieved for different orbital parameter combinations,

but correspond here to the values given in Table 3. For comparison, the present-day austral summer insolation (daily mean for DJF at 65˚S) is 439 $Wm^{-2}$. The eccentricity and the obliquity values resemble the cold orbital configuration, but the Earth is in perihelion during the austral summer today. The cold orbital configuration is chosen as the maximum insolation in an interval of ~40 kyr. This is about the time needed to grow a continental-scale ice sheet (Figure 12). The summer insolation is much

lower during shorter intervals of several millennia, but these intervals are too short to significantly increase the size of the Antarctic ice sheet and to induce a continental scale glaciation. Fig. 11 also indicates that a warm orbital configuration in Antarctica can either be caused by a high obliquity or a high eccentricity in combination with the Earth in perihelion. The highest peak in the summer insolation at 32.18 Ma occurs when both eccentricity and obliquity are high. The selected cold orbital configuration occurs at a time that the eccentricity reaches a minimum.


**Table 3: Austral summer insolation values, eccentricity and obliquity for the different selected orbital configurations. Present-day insolation values and orbital parameters (eccentricity and obliquity) are given for comparison.**

| Orbital configuration | Insolation (DJF-65˚S) | Eccentricity | Obliquity |
|---|---|---|---|
| 'cold' | 427 $Wm^{-2}$ | 0.017 | 22.5˚ |
| 'intermediate' | 440 $Wm^{-2}$ | 0.022 | 23.2˚ |
| 'warm' | 467 $Wm^{-2}$ | 0.049 | 23.3˚ |
| Present-day | 439 $Wm^{-2}$ | 0.017 | 23.4˚ |


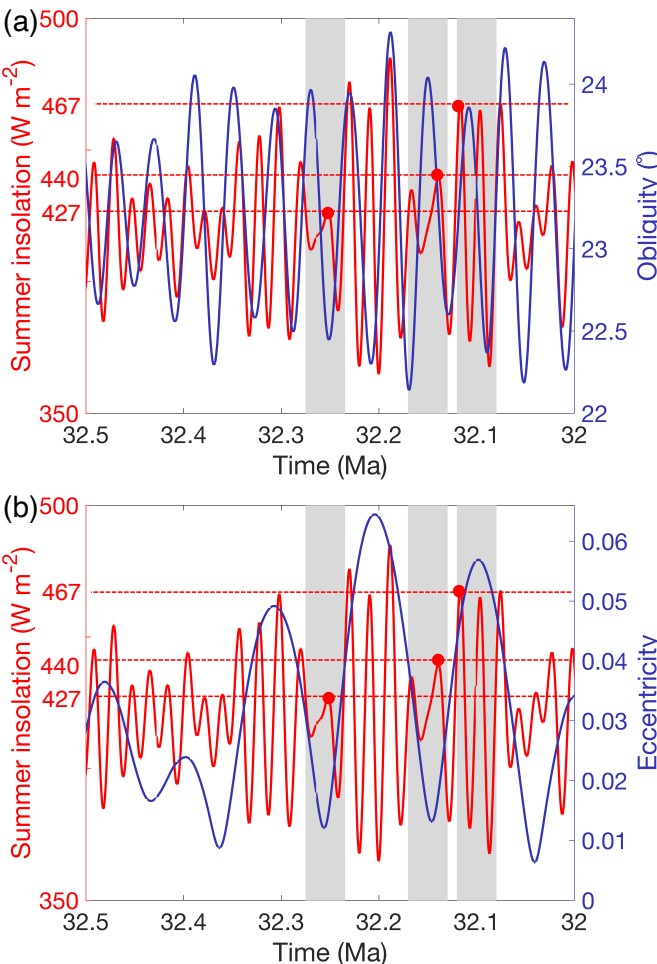

**Figure 11: Summer insolation variations at 65˚S with respect to the (a) obliquity and (b) eccentricity. Three different summer insolation values are indicated by the red dots that indicate the maximum insolation during a 40 kyr interval (indicated by the grey shaded area). These three different insolation values are representative for a relative cold orbital configuration (427 Wm⁻², an intermediate orbital configuration (440 Wm⁻²) and a warm orbital configuration (467 Wm⁻²).**

Starting from a bare bedrock, the Antarctic ice sheet can grow to a continental scale for our cold orbital configuration for a $CO_2$ forcing up to 850 ppmv (Figure 12a). For an intermediate orbital configuration, the ice sheet can grow to a continental scale up to a $CO_2$ forcing of 700 ppmv and for a warm orbit, the ice sheet is never able to grow beyond a critical threshold to induce continental scale ice sheet growth (not shown). Starting from a fully glaciated continent, the Antarctic ice sheet declines for a $CO_2$ forcing of 1000 ppmv and higher (Figure 12b). The ice sheet never completely deglaciates during a cold orbit or an intermediate orbit in the range of considered $CO_2$ concentrations (not shown).



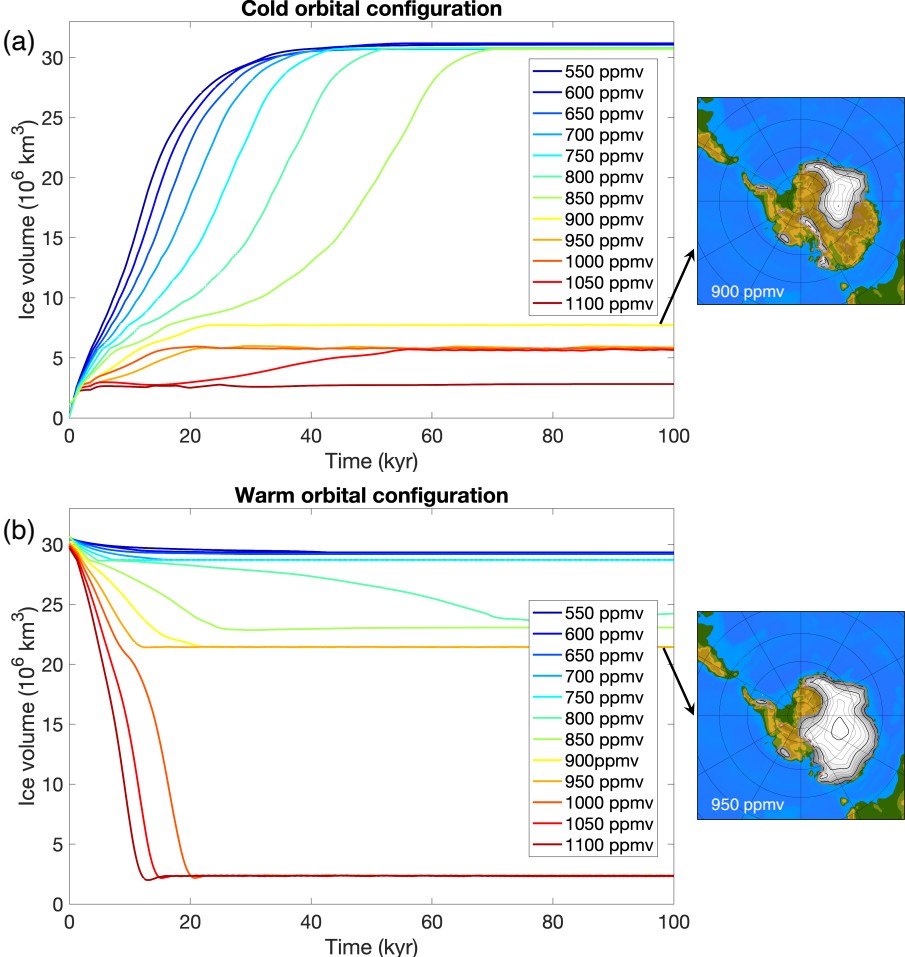

**Figure 12: Ice sheet evolution until steady-state for different constant forcing scenarios. (a) A cold orbital configuration (summer insolation at 65°S of 427 Wm$^{-2}$) and different $CO_2$ concentrations at interval of 50 ppmv starting from a bare bedrock. (b) A warm orbital configuration (summer insolation at 65°S of 467 Wm$^{-2}$) and different $CO_2$ concentrations at an interval of 50 ppmv starting from a fully glaciated continent. Ice sheet geometries are shown just before the tipping points are crossed at 900 ppmv for ice sheet growth (cold orbital configuration) and 950 ppmv for ice sheet decline (warm orbital configuration).**

The timescales for growth and decline of a continental ice sheet reveal an asymmetry. It takes about 30-70 kyr to build-up a continental scale Antarctic ice sheet, while the ice sheet demise takes about 10-20 kyr. The ice sheet grows much faster to a continental scale for a $CO_2$ forcing of 550 ppmv than for a $CO_2$ forcing of 850 ppmv, because the initial area of positive mass balance is much larger (Figure 13). A $CO_2$ concentration of 850 ppmv is close to the glaciation threshold and ice sheet-climate

feedbacks are necessary to make the ice sheet grow. Initially, the ice sheet grows fast up to 18 kyr when the ice sheet volume change stagnates and increases again after 30 kyr when the ice sheet covers about one third of the Antarctic continent and the ice-albedo and height-mass balance feedback reinforce the ice sheet growth. The highest mass balance rates occur along the ice sheet margin and range between 2-4 m yr$^{-1}$ accumulation. During the early stage of ice sheet build-up, the accumulation in the Gamburtsev Mountains goes up to 0.5 m yr$^{-1}$, while it is around 2 m yr$^{-1}$ in Dronning Maud Land. As the ice sheet grows, the accumulation lowers in central Antarctica, somehow similar to at present due to the development of a high-pressure area, the decrease in surface air temperatures and the depletion of its water vapour content, and the orographic precipitation along the margin that depletes the air moisture further.

The "coldest" orbital configuration, which corresponds to the lowest austral summer insolation occurs when the obliquity is low, the eccentricity is high, and the Earth is in aphelion. The time-dependent results clearly suggest that the eccentricity must largely determine the ice sheet growth because the time scale to initiate continental-scale glaciation is longer than the period of the precession or obliquity for the higher $CO_2$ concentrations. The $CO_2$ threshold to induce a glaciation in a transient setting is therefore lower for high values of the eccentricity. Fig. 14a shows the maximum ice sheet volume for simulations starting from a bare bedrock for a constant $CO_2$ concentration and a constant eccentricity (while precession and obliquity are varying; experiments (c) and (d) in Table 2). Fig. 14b shows the maximum ice sheet volume for simulations starting from a bare bedrock for a constant $CO_2$ concentration and a constant obliquity (while precession and eccentricity vary). Figs. 14c and 14d show the minimum ice sheet volume for similar runs, but starting from a continental scale ice sheet.

The higher the eccentricity, the lower is the $CO_2$ concentration to induce the deglaciation (Figure 14c). On the other hand, for a given $CO_2$ forcing below the glaciation threshold, the ice sheet can grow more for a higher eccentricity (Figure 14a). The higher eccentricity modulates the precession and for a high eccentricity the summer insolation minimum is lower when the Earth is in aphelion. As noted earlier, the duration of such a very low insolation minimum is not long enough to make the ice sheet grow on the entire Antarctic continent. Therefore, the $CO_2$ concentration needs to drop more to initiate ice sheet growth at high eccentricity values. Fig. 14 also illustrates this hysteresis behaviour, where the existence of a large ice sheet for a given forcing is dependent on the initial presence or absence of an ice sheet.

465

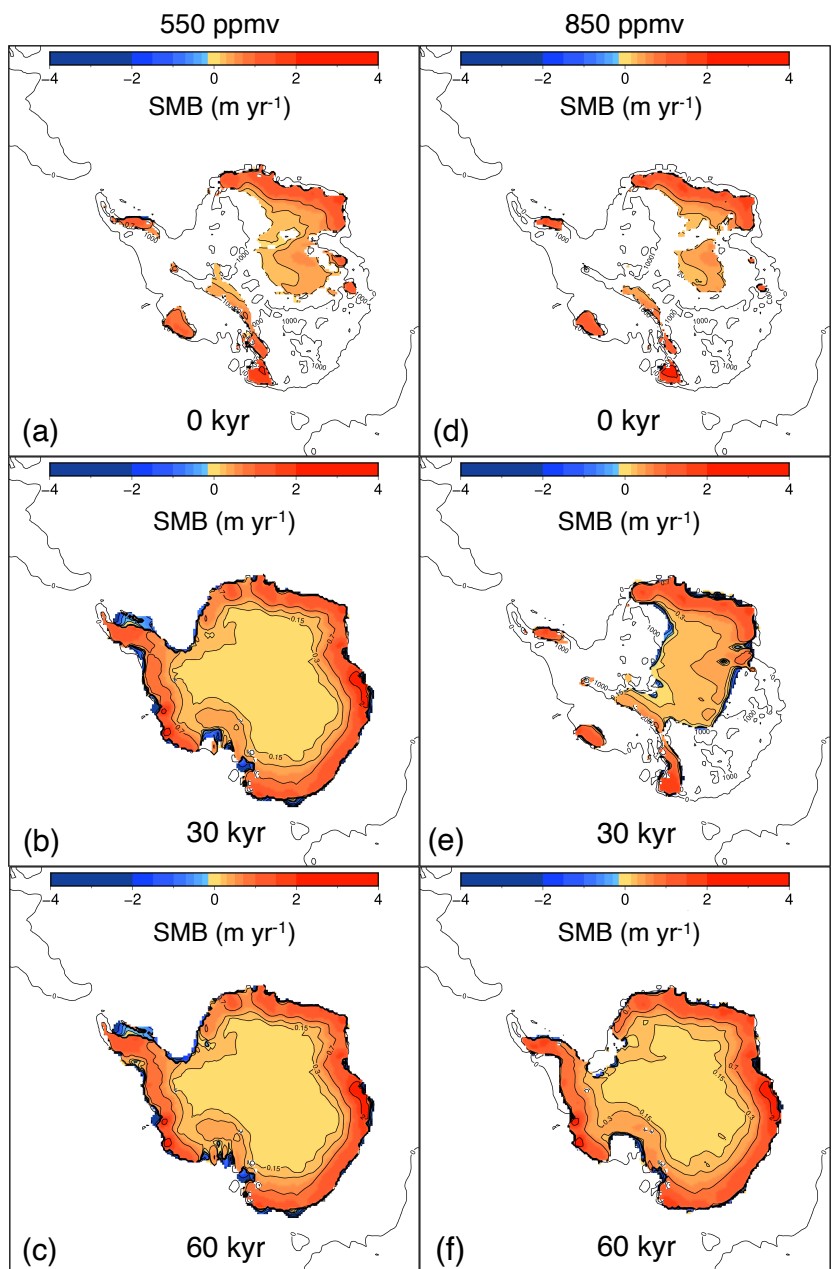

**Figure 13: Antarctic ice sheet surface mass balance (SMB) for the simulations starting from a bare bedrock forced by a cold orbital configuration and a CO₂ concentration of 550 ppmv after (a) 0 kyr, (b) 30 kyr, (c) 60 kyr and a CO₂ concentration of 850 ppmv after (d) 0 kyr, (e) 30 kyr, (f) 60 kyr.**

470

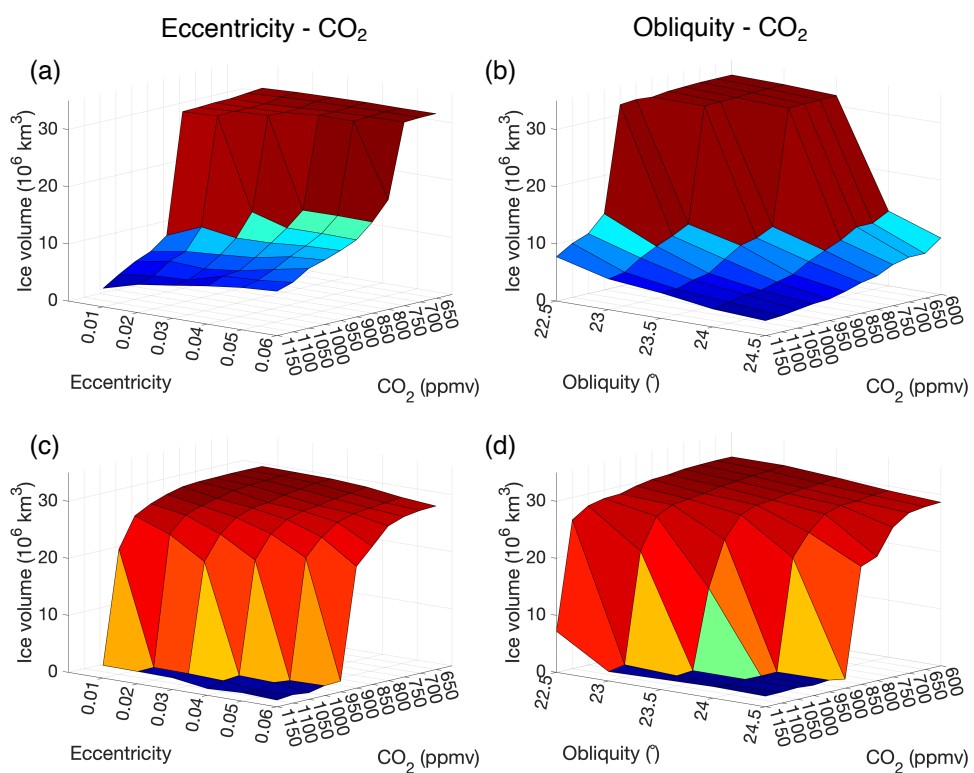

Eccentricity - CO₂ (a)

Obliquity - CO₂ (b)

(c)

(d)

**Figure 14: Simulations starting from bare bedrock showing the maximum ice sheet volume for a range in CO₂ concentrations between 600 ppmv and 1150 ppmv for (a) a constant eccentricity between 0.01 and 0.06, while precession and obliquity are variable and (b) a constant obliquity between 22.5° and 24.5°, while precession and eccentricity are variable. (c) and (d) are the same as (a) and (b), but the simulations start from a continental scale ice sheet and the minimum ice sheet volume achieved during the run is shown.**

Fig. 14b shows the maximum ice sheet volume for simulations starting from a bare bedrock for a constant obliquity and a constant CO₂ concentration, while Fig. 14d shows the minimum ice sheet volume for similar runs starting from a continental scale ice sheet (see experiments (e) and (f) in Table 2). The lower values for the obliquity clearly favour glaciation. For a constant obliquity of 22.5°, the ice sheet grows to a continental scale at 1000 ppmv, while for an obliquity of 24.5°, the ice sheet never reaches the continental scale. For the simulations starting from a continental scale ice sheet, the ice sheet never completely melts for a low obliquity of 22.5°, not even for a high CO₂ forcing of 1150 ppmv. This indicates that the role of the eccentricity for melting a large-scale ice sheet is only important during times that the obliquity is high. Looking at the transition between a glaciated and a deglaciated continent for a constant forcing, it is apparent that the ice sheet volume first gradually changes before a strong nonlinear feedback initiates. This effect is further investigated simulating steady state ice sheet geometries for a range in temperature anomalies starting again from either and ice-free continent (bare bedrock) and starting

from an ice sheet (Figure 15). Starting from the glaciated continent, first the ice sheet retreats with ~40% of the continental-scale ice sheet area before the non-linear feedbacks are initiated. On the other hand, when starting from a bare bedrock, the steady-state ice sheet area first gradually increases when lowering the surface temperature until the continent is covered with approximately 40% ice before a sharp transition occurs towards a fully glaciated continent.


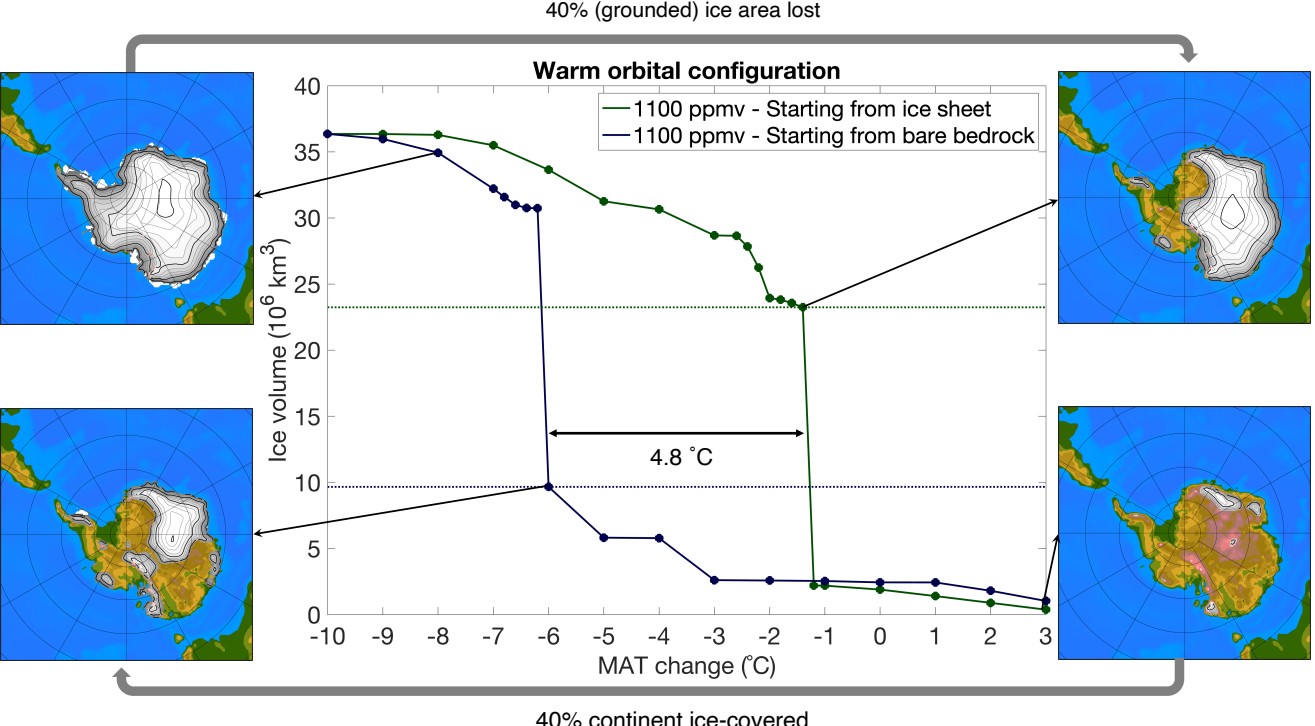

Figure 15: Steady-state ice sheet volume for a range in mean annual temperature perturbation thresholds for ice sheet growth and decline for a warm orbital configuration of 467 Wm$^{-2}$ at 65°S and a constant $CO_2$ concentration of 1100 ppmv. The green horizontal dotted line indicates the ice sheet volume at which the strong nonlinear ice sheet decline 500 initiates. The ice sheet volume at which a strong nonlinear increase ice sheet growth initiates is indicated by the blue horizontal dotted line. Snapshots of the ice sheet geometry are given at the tipping points. Thin contour intervals are given every 250 m, while thick contour intervals are given each 1000 m.

## 5 Discussion

Hysteresis behaviour related to the build-up and decline of ice sheets has a geometric cause (Oerlemans, 2002). The snowline 505 must lower to allow for a snow cover over the topography , initiating ice build-up. Obviously, the transition would be sharpest when the topography is flat and the snowline would intersect the entire topography at once. The lowering of the ice sheet

surface during decay causes the atmospheric temperatures to rise even more because of the lapse rate and accelerates the melting process, a process named the height-mass balance feedback. Because ice sheet hysteresis has a geometrical origin, the initial height of the bedrock topography and the height of the continental-scale ice sheet are crucial to determine the thresholds
for glaciation and deglaciation. We have confirmed that the $CO_2$ thresholds and especially temperature thresholds to initiate and terminate glaciations differ depending on the height of the bedrock. These $CO_2$ thresholds to initiate glaciations are 650 ppmv (minimum bedrock topography dataset) and 870 ppmv (maximum bedrock topography dataset). Aside from the bedrock dataset dependence, the thresholds also depend on the climate model used. As shown in Gasson et al. (2014), the thresholds may vary between 560 and 920 ppmv when climate model uncertainty is also included.


The hysteresis effect in our simulations has a magnitude of about 170-180 ppmv, slightly depending on the chosen bedrock topography dataset. $CO_2$ variations had a magnitude of at least 170-180 ppmv during the late Eocene, or about 70 ppmv larger than the 110 ppmv glacial-interglacial $CO_2$ variations of the Quaternary (Da et al., 2019). Given a sufficient low $CO_2$ level, this implies that Antarctic ice sheets could have waxed and waned also prior to the EOT. This was demonstrated in Van
Breedam et al. (2022), who used a $CO_2$ forcing based on available proxies that varied between 770 ppmv and > 1200 ppmv on a low temporal resolution to investigate ephemeral glaciations during the late Eocene. Such $CO_2$ variations are significantly larger than the hysteresis effect quantified in this study. One of the possible causes of the $CO_2$ variations during the early Cenozoic could be the build-up of the ice sheet itself, which causes a positive feedback loop with increased ice volume leading to lower $CO_2$ levels and amplified glaciations (Ruddiman, 2006).


The transition towards a glaciated state occurs as a sudden, nonlinear response when the ice-albedo feedback and height-mass balance feedback reinforce the ice sheet growth at the threshold value. It is suggested that albedo feedbacks between the ice sheet and the climate increase the strength of the hysteresis. Pollard and DeConto (2005) found a hysteresis effect of about 100 ppmv at the EOT, using a matrix look-up table with few initial ice sheet geometries that only roughly captured the ice-
albedo feedback. Their hysteresis curves also showed a more stepwise change towards full glaciation and deglaciation, mainly because the height-mass balance feedback strengthens the ice sheet growth each time the topography is above the snowline, leading to a (partly) glaciated continent.

Our simulations indicate that the ice sheet-climate feedbacks cause rapid ice sheet expansion when the ice sheet covers about
40% of the continent (Figure 15). Conversely, the Antarctic ice sheet area needs to decrease by about 40% (corresponding to an ice volume loss of 30% of the grounded ice volume) to initiate rapid ice sheet decay. This is in line with estimates from Ridley et al. (2010) who stated that the ice-albedo feedback becomes very effective when the ice sheet volume loss is about 30% for melting the present-day Greenland ice sheet. Mikolajewicz et al. (2007) found that a changing surface albedo when ice is melting increased the local temperatures by up to 10˚C in summer time. The sharp glacial-deglacial transitions are caused
by both the ice-albedo feedback and the height-mass balance feedback and it is thought that their impact on the surface

temperatures are of equal magnitude (Ridley et al., 2005). Opposite to the sharp transition between glacial and deglacial states from our study, studies that neglect the ice-albedo feedback such as Huybrechts (1994) and Garbe et al. (2020) show a more gradual transition into a deglaciated continent. Therefore, these studies might also overestimate the temperature forcing needed to melt the present-day Antarctic ice sheet.


Isostatic adjustment acts as a negative feedback during the glaciation of the Antarctic continent. Our simulations show that the strength of this feedback is quite large and lowers the $CO_2$ threshold to continental scale glaciations by about 130 ppmv. Oerlemans (2002) initially stated that the influence of isostatic movements should be small because the glaciation threshold occurs when the ice sheet is still small. However, our simulations indicate that the combined effect of a higher surface elevation

and more extensive albedo changes due to a larger area make a substantial difference in the glaciation threshold. During deglaciation, our simulations have indicated no significant impact of isostasy on the threshold to deglaciation. Isostatic uplift would reduce ice sheet melt, but the delayed response of the bedrock to changes in ice loading is too slow to counteract the increasing snowline (Abe-Ouchi et al., 2013). For the present-day Antarctic ice sheet, isostatic adjustment has a stabilizing effect on the Antarctic ice sheet evolution since at least the last 10 kyr (Kingslake et al., 2018). Also for future projections of

the Antarctic ice sheet on multi-centennial timescales, the uplift provides a negative feedback to future West Antarctic ice sheet loss by grounding line stabilization (Larour et al., 2019).

An interesting thought experiment is to assess the sensitivity of the early Cenozoic Antarctic ice sheet subject to the present-day orbital forcing (e = 0.0167, ε = 23.4393 and $\varpi$ = 102.9179; J2000; Laskar et al., 2004) and a pre-industrial $CO_2$

concentration of 280 ppmv. At present, the Earth is close to perihelion during the austral summer, a precondition to have high summer insolation values. Because of the low value of the eccentricity, the high summer insolation is attenuated and has a mean value for the austral summer at 65˚S of 439 $Wm^{-2}$. This is exactly in the middle between the selected "warm" and "cold" orbital configuration at the EOT. Imposing present-day orbital and $CO_2$ forcing leads to surface temperatures that are still up to 10˚C warmer around Dome A compared to the present-day Antarctic ice sheet (Figure S4). That is because the early

Cenozoic has a less poleward location than today. Prior to 22 Ma, the continental configuration also did not allow for the development of a strong circumpolar current (Evangelinos et al., 2022), but the full consequences of that are not assessed in the current setup of the climate model with only a mixed-layer ocean. The threshold to glaciation for a model run starting from a bare bedrock occurs at a temperature perturbation of +2.6˚C and the threshold to deglaciation occurs at a temperature perturbation of +7.6˚C (Figure S5). This is somewhat lower than the +10˚C found by Garbe et al. (2020) to melt the present-

day Antarctic ice sheet, on account of the different continental and climatic setting of the Eocene and processes taken into account (ice-albedo feedback). Also, in contrast to the present-day Antarctic ice sheet, the early Cenozoic Antarctic ice sheet is largely grounded above sea-level.

Non-linear ice sheet dynamics are also triggered by the orbital parameters. The eccentricity has a main influence on the initiation and termination of glaciations on the Antarctic continent with initiation during eccentricity minima and terminations during eccentricity maxima. This is perhaps puzzling at first because the eccentricity only influences the forcing by modulating the precession. However, because the ice sheet decline depends on crossing a threshold value, the amplitude of the precession cycle becomes important, and is governed by the eccentricity. Large values for the eccentricity result in a very low summer insolation during austral summer when the Earth is in aphelion and a very high summer insolation during austral summer when the Earth is in perihelion. To build a continental scale Antarctic ice sheet, the role of the eccentricity is important in order to keep the amplitude of the precession low. Abe-Ouchi and Blatter (1993) indicated already that the periods of precession (~20 kyr) and obliquity (~40 kyr) are too short to build-up large ice sheets when the accumulation rate is low and therefore, eccentricity has to be low to prevent the summer insolation to peak when the Earth is in perihelion.

Despite the performance of a substantial amount of sensitivity experiments related to the orbital and $CO_2$ forcing, we did not perform a sensitivity analysis of the ice sheet parameter uncertainty. Ice sheet model parameters such as the enhancement factor, the basal sliding coefficient or the flow rate factor each have their uncertainty and they are tuned to reproduce the present-day Antarctic ice sheet. Also the parameters related to the isostatic model are not explored. For instance, the relaxation time of the asthenosphere and the flexural rigidity of the lithosphere add another degree of freedom. The most important parameters influencing ice sheet-climate model feedbacks are however parameters related to the albedo (Gandy et al., 2023). Because our modelling approach on very long timescales significantly improved the representation of albedo changes resulting from changes in the ice sheet extent (Van Breedam et al., 2021a), we believe that the ice sheet parameter uncertainty is of secondary importance here.

**6 Conclusions**

We have shown that the early Cenozoic Antarctic ice sheet grew non-linearly during the late Eocene to Oligocene, when thresholds in the climate system were crossed. These thresholds at which the glaciation occurs depend on the boundary conditions such as the bedrock elevation. The $CO_2$ threshold to glaciation is ~650 ppmv for the minimum bedrock elevation and ~870 ppmv for the maximum bedrock elevation datasets used in this study. The hysteresis behaviour of ice sheets arises because of the positive feedbacks associated with ice sheet growth and decline such as the elevation-surface mass balance feedback and the ice-albedo feedback. The Antarctic ice sheet hysteresis effect is independent of the specific bedrock elevation dataset and is ~180 ppmv or equivalent to a 4°C to 5°C regional mean annual temperature change. Our simulations indicate the need to include ice-albedo feedbacks when the ice sheet is replaced by tundra during ice sheet melting and oppositely during ice sheet growth. The rapid transition between glacial and deglacial states as found in our simulations is attributed to this ice-albedo feedback and sets in when the Antarctic continent covers about 40% of the area (threshold to glaciation) or when the ice area decreases with 40% of the total ice area (threshold to deglaciation).

An important stabilising feedback is the isostatic depression of the bedrock when the ice sheet is building up. When the ice sheet is growing, the bedrock is depressed because of the weight of the overlying ice and the surface elevation is lowered. When this feedback is neglected, the ice sheet grows much faster to a continental scale and the hysteresis effect is significantly weaker. We found that this feedback is significant because of both an increased surface elevation and a larger ice sheet accumulation area that lowers the surface albedo.

The orbital parameters might regulate ice sheet growth and decline close to the $CO_2$ threshold for continental scale (de)glaciation. The role of eccentricity is especially important because ice sheet growth operates on a timescale of 30-70 kyr (longer than the precession and obliquity cycles) and ice sheet decline is initiated by crossing a threshold where the eccentricity determines the magnitude of this threshold value. The orbital parameters trigger both glaciations and deglaciations in a $CO_2$ range of ~80 ppmv, where the ice sheet develops rapidly when the ~100 kyr eccentricity cycle reaches a minimum (810-830 ppmv), the ~400 kyr eccentricity cycle reaches a minimum (850-880 ppmv) or the ~2.4 Myr eccentricity cycle reaches a minimum (890 ppmv) using the maximum bedrock topography dataset.

**Acknowledgement**

We thank two anonymous reviewers for their useful comments and very detailed feedback. Jonas Van Breedam acknowledges support from project G091820N, funded by the Research Foundation Flanders (FWO Vlaanderen).

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
