# Peer review of "Hysteresis and orbital pacing of the early Cenozoic Antarctic ice sheet"

_EGUsphere, 2023_

## Author Comment (AC1)

**Response to Anonymous Referee #1**

This study by Van Breedam et al. analyses the hystheresis of the Antarctic Ice Sheet during the Eocene-Oligocene Transition. They use an ice sheet model (AISMPALEO) forced by fields provided by an emulator (CLISEMv1.0), that in turn uses outputs from HadSM3 to produce such forcings (temperature and precipitation). By carrying experiments where oribtal parameters (or directly insolation in some cases), $CO_2$ levels, and the choice of bedrock topography are varied, they determine thresholds in $CO_2$ and eccentricity that are able to drive a complete deglaciation (glaciation) starting from a continental-scale ice sheet (ice free land).

They find a hystheresis behaviour for the ice sheet, where the difference between $CO_2$ levels necessary drive the ice sheet from a fully glaciated to an ice free state and vice versa is 170~180 ppmv. The absolute levels, however, vary according to the choice of prescribed bedrock topography, due to an interplay between bedrock elevation and snowline elevation. They further show that much of this hystheresis behaviour is due to the isostatic adjustment of the bedrock to ice (de)loading, as a fixed bedrock topography reduced this difference by more than a factor of 4. Finally, by performing tests under different constant $CO_2$ levels and varying orbital parametres, they find that eccentricity plays an important role in modulating the summer insolation values necessary to reach the threshold that will drive the ice sheet to a different state.

Overall, the quality of the work seems sound. The study offers good insights into the ice sheet hystheresis during a period where several geological changes were taking place (thus driving a substantial reorganisation of the climate system), and when several tipping points were crossed. Furthermore, it helps understand (and to a certain extent quantify) the drivers of permanent glaciation in Antarctica. I do think, however, that several technical aspects of the modelling approach need to be substantially clarified before the study's experimental design can be properly assessed and deemed publishable (which I think will be, don't get me wrong). Furthermore, some substantial reorganisation of the sections should be done, which would make it easier for the reader (and me as a reviewer) to follow the reasoning, methods used, and simulation ensembles carried out. These main concerns are outined below as "Major comments", while further suggestions to improve readability, as well as some minor corrections, are presented afterwards as "minor suggestions", "line by line comments", and "figure suggestions".

**Author's response:** We thank the reviewer for the in-depth review of the paper and the numerous suggestions, which has improved the manuscript significantly. In the revised version, we give more explanations about the technical components of the modelling study instead of relying too much on previous work, although we still believe that this description has to be brief and only serves to explain the main components, physics and parameterisations. Also, the order of some sections has been reconsidered, and where appropriate applied and many questions have been clarified.

Major comments

Several technical aspects of the model are missing or unclear, and need to be clarified in the paper. How many vertical levels does the ice sheet model have? Are they uniformly spaced, or refined closer to the base? What is the basal slding law used? How do you determine the

sliding coefficients? The dependence of ice deformation on its temperature is mentioned (L453-544), but there is no reference to how this interaction occurs in the model. How is ice deformation computed? Are any enhancement factors used? Is there any geothermal heat flux applied? Depending on how the parameters stated above are prescribed, they might strongly affect the results obtained. While I think performing extensive sensitivity tests to those parameters would make the paper even longer (and they are fairly unconstrained for the target period), the authors should discuss how their choice of model parameters might have influenced their results. Also, the ocean forcing seems to be completely ignored throughout the paper, except for a brief mention of marine-based ice sheets (L198-199). There is only a brief mention of ice shelves and the treatment of grounding lines, which is not clear at all (some of these concerns are listed further down in the line-by-line comments). How is basal melting of ice shelves prescribed? How is sea level prescribed? Even if the ocean plays a minor role in the EOT, this needs to be explicitly stated, and these aspects of the model still need to be described. From Fig. 6, there are indeed some ice shelves simulated. Finally, I would strongly suggest expanding more on how AISMPALEO and HadSM3 interact with CLISEMv1.0 (in section 2.3), including how/why it properly captures the ice-albedo feedback. As it is right now, it is not clear what is simulated and what is emulated. Since there's a lot of discussion on how the ice-albedo and elevation-mass balance feedbacks control the ice sheet response (and this is an important point of the study), it is crucial that the reader has at least a basic understanding of how the three model components interact without having to consult another publication.

**Author's response:** There are two lines of model technical aspects that have been clarified, the first one considers the ice sheet model AISMPALEO and the second part the ice sheet-climate model coupler CLISEMv1.0 which has been described in Van Breedam et al. (2021). Both models have been described before in extensive detail and therefore we do not explain everything again, but rather focus on the information needed for the understanding of the simulations performed here.

For the ice sheet model AISMPALEO description, we added the following information:

*L149-150:*
*The model has 30 levels in the vertical, with a closer spacing towards the bedrock where most of the shearing occurs.*

*L153-158:*
*The basal sliding velocity in AISMPALEO follows a Weertman relation and is proportional to the third power of the basal shear stress and inversely proportional to the height above buoyancy. The basal sliding coefficient is a constant multiplication factor for the basal sliding and equals $1.8 \times 10^{-10}$ $N^{-3}yr^{-1}m^8$. The sensitivity of the ice sheet model to ice sheet parameter uncertainties are not explored. An enhancement factor of 1.8 is used for grounded ice. This is similar to the value used to model the present-day Antarctic ice sheet. A constant geothermal heat flux of 50 mW $m^2$ has been applied over the entire model domain.*

*L160-167:*
*The grounding line is a one grid cell wide transition zone between the grounded and floating ice where all the stress components contribute in the effective stress in the flow law. Ice shelves develop when the grounded ice reaches the coast and the influx of snow from the atmosphere and ice from upstream exceeds the sum of surface ablation and basal melting. Although the slab ocean model exchanges heat with the atmosphere and records changes in the sea surface*

*temperature, we do not use this information to calculate basal melt rates. Instead, we prescribe a constant basal melt rate of 10 m yr$^{-1}$ over the entire domain. This is a strong simplification and perhaps it is even too low in some locations. It allows for small ice shelves to develop once the ice sheet reaches the coast. But even for the present-day, there is a large uncertainty in the way ocean water temperatures and salinity changes need to be translated into melt rates below the ice shelves.*

*L179-184:*
*The isostatic model consists of an elastic lithosphere with a flexural rigidity D of 10$^{25}$ Nm (which is a measure of the strength of the lithosphere) on top of a viscous asthenosphere, to allow the crust to deform far beyond the local ice loading (Huybrechts, 2002). The vertical deflection of the lithosphere w is given by a fourth order differential equation (Eq. 2) Here, q is the ice load, $\rho_m$ is the mantle density (3300 kg m$^{-3}$) and g the gravitational acceleration. This equation is solved using a Kelvin function of zero order (kei). The viscous asthenosphere responds to the ice load with a relaxation time τ of 3000 years (Le Meur and Huybrechts, 1996).*

$$D\nabla^4 = q - \rho_m g\, w \quad (2)$$

*L192-193:*
*The bedrock topography dataset assumes an ice-free continent. Sea-level changes (sea-level fall when the ice sheet is growing) are not included.*

For the emulator, only the precipitation and temperature patterns are emulated throughout the runs to drive the mass balance scheme of the ice sheet model. As such, the ice sheet model can run continuously, updating the forcing parameters and passing it to the emulator. The emulator returns the climatic fields for a given set of forcing parameters.

For the emulator CLISEMv1.0, we (slightly) extended on the albedo description as follows:

*L116-119:*
*The use of 20 different ice sheet geometries is equivalent to grasping the surface type differences at the resolution of the climate model and therefore the albedo changes are fully captured. The albedo varies between the discrete values of 0.8 for ice/snow and 0.2 for tundra.*

To be brief, the climatic fields (temperature and precipitation) are emulated to drive the ice sheet model. We rephrased the text as follows:

*L119-129:*
*After a calibration and validation process of the 100 preliminary climate model runs (see for details Van Breedam et al., 2021a), the emulator is able to provide the climatic forcing (temperature and precipitation fields) necessary to drive the mass balance of AISMPALEO for any combination of the orbital parameters, the CO$_2$ level and the ice sheet volume (Eq. 1). The orbital parameter combinations (Laskar et al., 2011) and the CO$_2$ concentration are prescribed, while the ice sheet volume ($V_{ice}$) is calculated within the ice sheet model.*

$$\begin{Bmatrix} T \\ precip \end{Bmatrix} = f(esin\widetilde{\omega}, ecos\widetilde{\omega}, \varepsilon, CO_2, V_{ice}) \quad (1)$$

The experimental design for the ice sheet model comes much later than the model description, and several experiments are only mentioned down the line. I would suggest to change the order where the models are presented in section 2 (e.g., leaving the ice sheet model description as 2.3, right before delving into the different experiments), and clearly state all experiments shown in sections 3 and 4 already in section 2. The way the experiments are presented in the manuscript right now is too scattered, making it hard to follow. A good way to solve this issue would be by having a single summarising table, with the experiments grouped by "goals". This would make it easier for the reader to have an overview of what has been tested (as opposed to having 3 tables - are the experiments in Figs. 12-15 in them? this is not clear). In that way, it would be easy to refer to this table (and to which group of simulations is being evaluated) throughout the different sections, so the reader can easily follow which parameters are being kept fixed, and which are being varied (especially when changes in CO2 and orbital parametres are being discussed). This might require a substantial ammount of rewriting of some sections, but it will greatly improve the flow of the manuscript.

**Author's response:** It is not the experimental design for the ice sheet model simulations, but the experimental design for the coupled simulations. In that sense, the experimental design follows immediately after the description of the different model components.

Since there is no need to start with the ice sheet model description, we changed the order and start now with the description of the climate model, then the emulator and finally the ice sheet model before we dive further into the experimental design of the coupled ice sheet-climate simulations.

The discussion is very concise and easy to follow, but I miss a bit more of detail on what these thresholds mean to the current state of Antarctica, if anything at all. If not, in what do they differ so that such high CO2 levels can still sustain continental-scale ice sheets? One thing that strikes me is the the fact that no attention has been given to the ocean at all, both in the description of the model (already stated above) and when discussing the results. Is that because there are barely any ice shelves? If that is the case, why? Could this be because of how the model simulates them? The technical aspects of the model need to be clarified (as stated above). If ice shelves are present and play some role, how much would the fact that the climate component uses a slab-ocean model affect the results?

**Author's response:** We are happy to hear the concern about the thresholds for the current state of Antarctica. Initially, we wanted to include a comparison with the current state of the Antarctic ice sheet. But, the climate during the late Eocene and the early Oligocene was completely different than the climate we have today or even future climatic changes. We performed additional runs where the early Cenozoic Antarctic ice sheet is simulated for a present-day forcing (orbital and pre-industrial $CO_2$ forcing) to illustrate the different climatic setting.

The reason why little attention has been given to the ocean is the fact that in most simulations almost all of the ice sheet is resting on land above sea-level. The palaeotopography around 38-33 Ma ago was much higher than it is today, especially for West Antarctica. Even for a continental-scale ice sheet that has depressed the underlying bedrock, the ice is still resting on a bed largely above sea-level. Therefore, the marine ice sheet dynamics had a minor influence on ice sheet retreat.  However, in some place ice shelves do develop and we added the following text as discussion:

*L162-167:*

*Although the slab ocean model exchanges heat with the atmosphere and records changes in the sea surface temperature, we do not use this information to calculate basal melt rates and instead, prescribe a constant basal melt rate of 10 m yr⁻¹ over the entire domain. This is a strong simplification (and perhaps not enough in some locations) and allows for small ice shelves to develop once the ice sheet reaches the continental size. But even for the present-day, there is a large uncertainty in the way ocean water temperatures and salinity changes need to be translated into melt rates below the ice shelves.*

Minor suggestions

- Although potential tipping-point triggers for the EOT are mentioned early in the introduction, it is not clear from the start why this study focuses on this period. The reasoning is only made explicit at the very last sentence of the introduction. It would be good to highlight much earlier why this period was chosen, so the reader can be more engaged.

**Author's response:** We thank the reviewer for this suggestion and added the following lines in the second paragraph of the introduction:

*L30-34:*

*Geological evidence (Scher et al., 2014; Carter et al., 2017) and modelling work (Van Breedam et al., 2022) also pointed to ephemeral glaciations prior to the EOT. This would imply that thresholds in the climate system were first crossed to initiate and end large-scale glaciations during the late Eocene.*

- The description of hystheresis in L42-47 and the definition stated L442 are not quite the same. How can they be reconciled?

**Author's response:** We do not see where the definition is given on L442.

- For an easier reading, I strongly suggest to explicitly say "mean surface temperatures" as opposed to "MAT_sur". Similarly for "MAT_clim"

**Author's response:** The reason why we chose for MAT_sur and MAT_clim is to avoid confusion between the climatological mean annual temperature and the mean annual temperature at the surface that varies because of surface elevation changes. We believe it adds clarity to the different variables used.

- The solid Earth interaction with the ice sheet is defined in several different ways. "solid Earth rebound (feedback)", "isostatic adjustment (feedback)", "isostatic rebound", "isostasy". Please stick to one way for consistency, unless appropriate for that specific context/explanation.

**Author's response:** We agree with the reviewer that too many different terms for the same process were present in the manuscript. We avoided the term 'solid Earth rebound/feedback' and either used isostasy (process) or isostatic adjustment (feedback).

Line by line comments

L62: "deform" is a more appropriate term than "deflect"

**Author's response:** Corrected.

L67: It is not clear what is meant with "surface type". Is it ice/bare rock/snow? It needs to be clarified.

**Author's response:** We added the surface type here as follows:

*L70-71:*

*This number of different ice sheet geometries allows the climate model to represent the climatic state for a small change in the surface type, being either ice or tundra.*

*L118-119:*

*The albedo varies between the discrete values of 0.8 for ice/snow and 0.2 for tundra.*

L69: "potentially significant impact"

**Author's response:** Corrected.

L70-73: What would be a "constant curve"? In the following sentence, it is not clear which parameters are kept constant and which are not. Do you mean that you vary 2 orbital parameters while CO2 and a third orbital parameter is kept constant? Or you vary one orbital parameter and CO2? It is quite hard to follow.

**Author's response:** Here we summarize the main goals and experiments performed. The full experimental design is discussed in section 2.4. To avoid confusion, we rephrased some of the sentences.

*L75-77:*

*Additionally, the importance of the orbital parameters on the glaciation and deglaciation thresholds is investigated. Constant forcing simulations are run to explore the influence of the eccentricity, the obliquity and the $CO_2$ values on glaciation and deglaciation thresholds.*

L89-90: Do you mean that you are solving for the full-stokes equations, or a combination of SSA/SIA? I assume the "transition zone" is the grounding line, but this needs to be explicitly described.

**Author's response:** The transition zone is indeed the grounding line. We rephrased the text as follows:

*L160-161:*

*The grounding line is a one grid cell wide transition zone between the grounded and floating ice where all the stress components contribute in the effective stress in the flow law.*

L90-91: Does that mean that you have a fixed mask to determine "the coast"? Or how is that defined?

**Author's response:** The coast is dynamically defined a floatation criterion based on ice thickness and bedrock elevation. If there is no ice at the coast, the coastline follows the zero-elevation contour.

L91: by mass balance I assume you refer to surface mass balance? Please update accordingly

**Author's response:** Corrected.

L91 and 94: PPD and DDF are acronyms that are not defined. Not every reader of the journal is familiar with glaciology technical terms, so these should be explicitly defined.

**Author's response:** Corrected.

L95: As in the comment above, please define "m i.e." the first time it is used. It might be straightforward for the glaciological community, but CP is a journal that targets a wider audience, and therefore not everybody would be used to these more specific units

**Author's response:** Corrected.

L98-100: What are the values for the lithosphere rigidity and viscosity? All values should be clearly stated somewhere, so that results can be reproducible

**Author's response:** We extended the explanations on the GIA model since it is an important component regarding the isostasy sensitivity runs.

*L179-187:*

*The isostatic model consists of an elastic lithosphere with a flexural rigidity D of $10^{25}$ Nm (which is a measure of the strength of the lithosphere) on top of a viscous asthenosphere, to allow the crust to deform far beyond the local ice loading (Huybrechts, 2002). The vertical deflection of the lithosphere w is given by a fourth order differential equation (Eq. 2) Here, q is the ice load, $\rho_m$ is the mantle density (3300 kg $m^{-3}$) and g the gravitational acceleration. This equation is solved using a Green's function. The viscous asthenosphere responds to the ice load with a relaxation time τ of 3000 years (Le Meur and Huybrechts, 1996).*

$$D\nabla^4 = q - \rho_m g \, w \quad (2)$$

L107: What anomalous heat convergence? Due to what? This really comes out of the blue here

**Author's response:** In slab ocean-atmosphere models, an anomalous heat convergence is added to the ocean in order to mimic the real oceanic circulation. The slab (50 m in our simulations) is still exchanging heat with the atmosphere, but the deep ocean is not. We believe the explanation that is present in the next sentence is sufficient to understand the influence of the anomalous heat convergence.

L111-112: what is the resolution of the bedrock dataset? This might not be the appropriate section to mention it. As far as I understand, this is part of the ice sheet model setup, not the climate model.

**Author's response:** We added some more explanation on the bedrock dataset and how we created it for the ice sheet model simulations:

*L98-103:*

*The bedrock topography used in the climate model is the Wilson maximum bedrock topography and is representative for the Eocene-Oligocene transition (EOT) at 34 Ma (Figure 2). The minimum and maximum bedrock topographies are applied as a boundary condition in the ice sheet model at a 40 km resolution. In order to grasp the entire uncertainty, each ice sheet model grid cell takes on the lowest and highest value for respectively the minimum and maximum bedrock topography from the original higher resolution Wilson et al. (2012) dataset within each ice sheet grid cell.*

L120-122: How were these initial geometries created? Are they also the ones used as initial conditions for the hystheresis experiments?

**Author's response:** These initial conditions were created based on steady-state simulations ranging from high forcing to low forcing scenarios to have a good spread between minimal ice coverage and maximum ice coverage on the continent. The ice sheet simulations always start from minimal ice sheet coverage (nearly bare bedrock) or from the reverse run when the ice sheet reached its maximum extent (so not from these initial geometries).

L123: does the ice sheet model not receive any ocean field?

**Author's response:** No, we chose to not use the temperatures from the ocean model to infer a basal melt parameterisation for various reasons. First of all, this is not relevant for ice sheet initiation. Secondly, even for the present-day, it is highly uncertain how the ocean temperatures should be translated into basal melt rates. Thirdly, the water temperatures are only calculated in the 50 m slab ocean layer. The basal melt rate is usually parameterised as a function of the depth and therefore, the model does not provide the necessary information to infer a spatially variable basal melt rate.

L126: "every 500 years" reads better

**Author's response:** Corrected.

L134: "only ice" -> "ice is present only"?

**Author's response:** Thank you for the suggestion, we rephrased the sentence as follows:

*L134-135:*

*The simulated Antarctic climate is strongly dependent on the forcing. In Figure 3, seasonal temperature and precipitation patterns are illustrated for a nearly ice-free Antarctica having a few small ice caps on the highest elevations only.*

L137: "land inwards" -> inland?

**Author's response:** Corrected.

L133-L142: reads like results?

**Author's response:** We believe the temperature fields belong to the model description, as a visualisation of the model performance and the resolution of the climate model.

L147-149: I find it curious that the ice sheet grows to a continental scale at 550 ppmv. What were the initial conditions for this ice sheet growth experiment? As the paper highlights, the initial conditions are quite important for the resulting ice sheet (see comment above on L120-122)

**Author's response:** The glaciation threshold is somewhere between 670 and 870 ppmv, depending on the applied bedrock topography dataset. Hence, the ice sheet grows always to a continental scale for a $CO_2$ forcing of 550 ppmv.

L149: "ppm" is used, as opposed to ppmv, which is the one used throughout most of the paper.

**Author's response:** Corrected.

L155: By "control parameters", do you mean that they were kept constant?

**Author's response:** Yes. We rephrased the sentence:

*L202-204:*

*In these runs, different values for the individual orbital parameters or the insolation are explored and the $CO_2$ concentrations are kept constant.*

L156: What are these values?

**Author's response:** These values are given in Table 2.

L160: You state in L144 that all experiments span 10 Myr, yet here you state that some are only 2.4 Myr long. It would be easier to follow the description if the experiment duration was added to the table of experiments as well.

**Author's response:** What we mean is that all experiments take place between 30 and 40 Ma. Some are 10 Myr long (the bedrock sensitivity experiments), others where the time-dependent orbital parameters are investigated are 2.4 Myr long and the constant forcing hysteresis experiments are only 200 kyr long. The experiment duration has been added to Table 2. A short description of the experiment duration and the $CO_2$ window for the constant $CO_2$ experiments has been added:

*L207-213:*

*The duration of the experiments where the influence of the eccentricity is investigated is 2.4 Myr long starting at 34.2 Ma up to 31.8 Ma, to capture the extrema when the 100 kyr, 405 kyr and 2.4 Ma cycles reach a maximum, separately and combined. In these experiments, the $CO_2$ forcing is explored in a narrow window of 80 ppmv at an interval of 10 ppmv. The other experiments where the eccentricity is constant and the obliquity is variable or the obliquity is constant and the eccentricity is variable are sampled in a larger $CO_2$ window of 450 ppmv at an interval of 50 ppmv (Table 2). They have a duration of 200 kyr because they equilibrate faster to the forcing (there is a limited influence of changes in the orbital forcing).*

L165: "minimum and maximum topography estimates"

**Author's response:** Corrected.

L167: "linearly decreased" - but from which value to whch value?

**Author's response:** This information was already given (L145-146). We removed some information from L167 and added the remaining text to the first paragraph of section 2.4.

L168: This (partly) answers my question raised above for L120-122! I think some reorganisation of the experimental design and methodology would make the experimental design much easier to follow

**Author's response:** We have moved this section to the start of the experimental design.

L180: "In section 3.1 we test the sensitivity of the Antarctic ice sheet hysteresis to the bedrock topography dataset" reads much better

**Author's response:** We changed the sentence to:

*L225-226:*

*In section 3.1 we test the sensitivity of the Antarctic ice sheet hysteresis to the initial bedrock topography dataset.*

L184: "chosen" or "prescribed" would fit better than "applied".

**Author's response:** We changed "applied" to "prescribed".

L189-190: I am not sure I understand the use of "except" in this sentence

**Author's response:** We removed that sentence and added the information to the previous sentence as follows:

*L234:*

*The higher the initial bedrock topography, the larger the final extent and elevation of the ice sheet.*

L191: what do you mean by "lower above the ice sheet because the elevation is higher"? Are you not talking about the surface temperature? Or do you mean further inland?

**Author's response:** We are talking about the mean annual surface temperature over the entire continent. We changed "because of the higher elevation" to "because of the higher mean elevation".

L226: this is the first mention of "rebounded topography". Is it because you are referring to the deglaciated runs? If so, it should also be stated for the minimum bedrock topography as well. If not, does it mean that the other deglaciation experiments mentioned had no isostatic model included?

**Author's response:** We removed the word "rebounded" to avoid confusion. Anyway, all these experiments described here include isostatic rebound. So in case the continent deglaciates, you could think of a rebounded bedrock topography. But anyway, that is the same as the original bedrock topography dataset when you wait long enough.

L228: I would suggest to use another word rather than "march", so that it is not mistaken with the month of March. One alternative is to use "range" instead

**Author's response:** Given the context and the different spelling (capital letter), we think there could be no confusion.

L231 and 232 (and throughout): maximum and minimum topographies, as you are referring to more than one

**Author's response:** Corrected.

L243: I suggest use "deform" instead of "deflect" again

**Author's response:** Corrected.

L243-244: I am not sure I follow the reasoning behind the lowering of the snowline being delayed. The atmospheric cooling is the responsible for lowering the snowline, and with a delayed increase in ice sheet elevation, what happens is that it will take longer for the same extent of the ice sheet to be above the snowline. These are not quite the same thing (relative vs. absolute points of reference)

**Author's response:** We rephrased this sentence as follows:

*L289-290:*

*Hence, the ice sheet elevation will not rise as fast and the increase in accumulation area (when the ice sheet's height exceeds the snowline) will be delayed.*

L244-245: Again, new experiments are being introduced... Please summarise all in a single table as stated in the major comments.

**Author's response:** We now summarized all experiments in Table 1 and Table 2 of the experimental design. Therefore, we added two columns to the tables: one that indicates whether isostasy is included and another one that indicates the duration of the experiments.

*Additional sensitivity experiments are performed where the isostasy feedback is not taken into account.*

*Table 1: Standard set of experiments with the Wilson minimum and Wilson maximum bedrock topography as boundary conditions and variable orbital forcing.*

| | Ice level (at start) | CO$_2$ (ppmv) | Eccentricity | Obliquity | Isostatic adjustment | Experiment duration |
|---|---|---|---|---|---|---|
| Wilson minimum | No ice | 1150 to 550 (linear) | variable | variable | Yes | 10 Myr |
| Wilson minimum | Ice | 550 to 1150 (linear) | variable | variable | Yes | 10 Myr |
| Wilson maximum | No ice | 1150 to 550 (linear) | variable | variable | Yes | 10 Myr |
| Wilson maximum | Ice | 550 to 1150 (linear) | variable | variable | Yes | 10 Myr |
| Wilson maximum | No ice | 1150 to 550 (linear) | variable | variable | No | 10 Myr |
| Wilson maximum | Ice | 550 to 1150 (linear) | variable | variable | No | 10 Myr |

*Table 2: Experiment overview for the runs investigating the influence of the orbital parameters for fixed CO$_2$ concentration levels.*

| | Ice level (at start) | CO$_2$ (ppmv) | Eccentricity | Obliquity | Isostatic adjustment | Experiment duration |
|---|---|---|---|---|---|---|
| **Wilson maximum** | Ice | 980, 990, 1000, 1010, 1020, 1030, 1040, 1050, 1060 | Variable | Variable | Yes | 2.4 Myr |
| **Wilson maximum** | No ice | 810, 820, 830, 840, 850, 860, 870, 880, 890 | Variable | Variable | Yes | 2.4 Myr |
| **Wilson maximum** | Ice | 650, 700, 750, 800, 850, 900, 950, 1000, 1050, 1100, 1150 | 0.01, 0.02, 0.03, 0.04, 0.05, 0.06 | variable | Yes | 200 kyr |
| **Wilson maximum** | No ice | 650, 700, 750, 800, 850, 900, 950, 1000, 1050, 1100, 1150 | 0.01, 0.02, 0.03, 0.04, 0.05, 0.06 | variable | Yes | 200 kyr |
| **Wilson maximum** | Ice | 600, 650, 700, 750, 800, 850, 900, 950, 1000, 1050, 1100, 1150 | variable | 22.5°, 23°, 23.5°, 24°, 24.5° | Yes | 200 kyr |

| Wilson maximum | No ice | 600, 650, 700, 750, 800, 850, 900, 950, 1000, 1050, 1100, 1150 | variable | 22.5°, 23°, 23.5°, 24°, 24.5° | Yes | 200 kyr |
|---|---|---|---|---|---|---|

L279: the use of "or" here is confusing. If you are defining/naming MAT_clim as the mean temperature at sea level, state explicitly that this is how you are defining it. Or are they two completely different things?

**Author's response:** No they are equivalent and we explicitly mention that now:

*L322-324:*

*To disentangle the influence of surface elevation, the MAT_sur is now corrected for the surface elevation change by applying a constant lapse rate correction in order to calculate the temperature at sea-level, which we equate to the climatological mean annual temperature (MAT_clim).*

L281-283: The reasoning here is not quite clear. I would suggest you more explicitly explain it as in L283-285 for the initial difference in temperature

**Author's response:** We rephrased the text as follows:

*L324-328:*

*As it occurs, the initial temperature difference of about 0.5˚C between both experiments with $CO_2$ concentrations between 1000 and 1150 ppmv is due to the larger ice sheet area in the experiment that excludes isostasy. After the transition to a continental scale ice sheet for a $CO_2$ concentration below 850 ppmv, this difference between both simulations is negligible because the area of the continental scale ice sheet becomes nearly the same as the ice sheet extent and is ultimately bounded by the size of the Antarctic continent.*

L294: This sentence does not read well. You can start with either "there is a CO2 threshold to..." or "A CO2 threshold to continental-scale glaciation exists". Also, "for a certain" makes me wonder if you are talking about your experiments or not. I suggest "for each tested bedrock topography" instead. If that is what you mean, it might be good to highlight in the sentence that these thresholds are different for each of the tested bedrock topographies.

**Author's response:** The sentence has been rephrased:

*L339:*

*There is a different $CO_2$ threshold to continental-scale glaciation for each tested bedrock topography.*

L297-299: I am not sure I follow the explanation here. How does the time indicate the change in sensitivity of the ice sheet? do you mean the time when the tipping point is crossed? In the panretheses, are you stating that in each experiment the initial conditions are representative of

the ice sheet at a certain point in time (in which case it would have been through a different number of precession cycles)? This bit needs to be rewritten for clarity.

**Author's response:** In the experimental design, it is now clearly mentioned that these experiments last 2.4 Myr or a long eccentricity cycle. The experiments start all with the same initial conditions, but throughout the runs the ice sheet is changing size and therefore, the conditions at an insolation minimum are different throughout the run. We rewrote the text as follows:

*L342-346:*

*All simulations start with the same initial conditions and the ice sheet volume responds to the orbital forcing. Throughout the runs, the ice sheet geometry has changed at each eccentricity extremum (~100 kyr, ~400 kyr, ~2.4 Myr) because the ice sheet size reacts periodically to the precession (~20 kyr). The time axis indicates the duration since the start of the simulations to reach the eccentricity threshold to glaciation or deglaciation.*

L304-305: This is a very interesting statement, and would be good to have a figure to illustrate it. It would be of great support to understanding Fig. 10 (maybe as a second panel?)

**Author's response:** We added an additional figure (FigureS3) in the supplementary information showing the time series of the run to initiate glaciations and deglaciations.

L317: I find it odd to use "remarkable" here. Isn't that exactly what hystheresis is about, as you state later in L442? Also, the dependence of the ice sheet response on initial conditions (both geometry and thermal state) is something quite actively discussed and not a surprise.

**Author's response:** We removed the word remarkable and moved this section to the first paragraph of section 4.1.

L351-352: isn't that true for the 'intermediate' case as well?

**Author's response:** No, the 'intermediate' orbital configuration has a high obliquity, but a low to medium value for the eccentricity.

L375: While the 40kyr time presented in L346-347 is within the bounds stated here, it makes me wonder what configuration yields the 40 kyr time used as justification in L346-347? It would be good to state that for clarity when comparing these two statements

**Author's response:** We refer on L346-347 already to Figure 12, where it can be seen that after 40 kyr, the ice sheet has grown to a continental-scale, except for higher $CO_2$ values close to the glaciation threshold.

L384: Is that only due to the high pressure area? The high pressure area at the poles exists even in the most idealised (e.g., "water planet") Earth atmopsheric circulation scenarios. I suggest it might also be due to an orographic effect caused by the ice sheet growth itself, which acts as a barrier to weather systems and restricts them to the margin (where most accumulation occurs).

**Author's response:** There are definitely other reasons why the precipitation decreases as the ice sheet develops. Probably most importantly because the temperature gets lower as the ice sheets develops and the resulting depletion of water vapour for decreasing air temperatures.

*L439-442:*

*As the ice sheet grows, the accumulation lowers in central Antarctica, somehow similar to today due to the development of a high-pressure area, the decrease in surface air temperatures and the depletion of its water vapour content, and the orographic precipitation along the margin that depletes the air moisture further.*

L399: I believe you meant "as noted earlier"

**Author's response:** Corrected.

L399-401: I suggest breaking up this sentence in two for an easier flow

**Author's response:** Done.

L401: I suggest rewriting to "Fig. 14 also illustrates this hystheresis behaviour...". As it is now, it's as if the figure itself had some type of hystheresis...

**Author's response:** Done.

L429: This sentence construction is weird. I suspect you meant "either an ice-free continent (bare bedrock) or from an ice sheet"

**Author's response:** Corrected.

L432: "gradually increases when lowering..."

**Author's response:** Corrected.

L443: This sentence is hard to follow, with "to allow", "to develop", and "to initiate" one after the other. I suggest breaking up or rephrasing as "... topography and initiate ice build up" or alternatively "... topography, initiating ice build up"

**Author's response:** This sentence has been rephrased.

L446: I would refrain from using terms such as "very" or "very much". You state the same thing without including them here, the study already demonstrates that.

**Author's response:** Done.

L451: I would state "depending on the choice of/chosen bedrock topography" instead, to avoid the reading mistaking it for a local topographic dependency. I would also remove "slightly", as the experiments demonstrate it is exactly due to the different topographies.

**Author's response:** We rephrased the sentence but left the word slightly, because the hysteresis effect is only slightly dependent on the chosen bedrock topography dataset. The

thresholds are very different but the hysteresis effect is either 170 ppmv (minimum bedrock topography) or 180 ppmv (maximum bedrock topography).

L457: "This is mainly because"

**Author's response:** Corrected.

L461: "Conversely" reads better than "oppositely"

**Author's response:** Corrected.

L485-486: For an easier reading, I suggest rephrasing as "the amplitude of the precession cycle becomes important, and is governed by the eccentricity."

**Author's response:** Done.

L488: I suggest rephrasing as "important in order to keep"

**Author's response:** Done.

L493: This is a bold statement, as these results are based only the model results from this study. It is more honest to state that this is what your model/experiments show, especially because there has been no comparison to geological constraints, neither an assessment of the robustness of the ice sheet model to unconstrained model parametres.

**Author's response:** We have rephrased the sentence to:

*L580:*

*We have shown that the early Cenozoic Antarctic ice sheet grew non-linearly during the late Eocene to Oligocene, when thresholds in the climate system were crossed.*

L494: "these thresholds [...] depend" - correct for plural

**Author's response:** Done.

L496: why is "ice sheets" in plural? So far the AIS has been treated as a single ice sheet

**Author's response:** Because the hysteresis behaviour is not solely an effect that is specific for the Antarctic ice sheet, but for ice sheets in general.

L512: I suggest rephrasing to "The role of eccentricity is especially important" for easier reading

**Author's response:** Corrected.

L515: I suspect you mean 180 ppmv? This is the first time the number 80 comes in the story, and is at the closing sentence of the paper!

**Author's response:** No, the 80 ppmv refers to the window in which the eccentricity is pacing glaciations and deglaciations. We mention now in the experimental design that these experiments span a CO2 window of 80 ppmv at an interval of 10 ppmv.

References

Baatsen et al. (2020) is cited, but not in the references

Payne et al. (2005) is in the references but not cited

Zeitz et al. (2021) is in the references but not cited

**Author's response:** We thank the reviewer for the detailed checks and corrected the references.

There might be other references that are cited but are not listed in the references. I would strongly advise the use of a reference manager to help keeping track of the citations.

**Author's response:** Thank you for noticing. The reference list has been updated.

Figure comments

Figure 2: a thrid panel showing the difference between both topographies would be beneficial to highlight in which areas they differ the most (and how uniform that difference is). This would help quite a bit when discussing the effect of the chosen bedrock topography and the mass balance-elevation feedback

**Author's response:** We added a third panel showing the difference between the maximum and minimum bedrock topographies. Also, we add some more explanations on how the maximum and minimum bedrock topographies have been reconstructed and used in the ice sheet model simulations.

Figure 3: I think it is confusing to mention 2x Pre-Industrial CO2 in the caption. Since you discuss all results in terms of the absolute values, it would make more sense to state 560 ppmv instead.

**Author's response:** We left the 2xpre-industrial forcing and added between brackets 560 ppmv. Many model runs are performed for these idealized forcings (2xPI, 4XPI) and therefore we would like to name it as such.

*Antarctic climatologies for a 2x pre-industrial $CO_2$ forcing (560 ppmv)*

Figure 4: What was the window used for the running mean mentioned in the caption? The period between 34.2 and 31.8 is very hard to see in the figure. I suggest adding a background shaded box to properly highlight it.

**Author's response:** Thank you for the good suggestion. We added a grey shaded region to highlight the period between 34.2 and 31.8 Ma.

Figure 6: what elevation to the contour lines represent in panels (a) and (c)?

**Author's response:** We added the contour line interval information.

*Thin contour intervals are given every 250 m, while thick contour intervals are given each 1000 m for the ice sheet surface elevation.*

Figure 10: I do not understand what you meant with the statement on the experiment duration. Are you using the resulting geometry as initial conditions to another experiment? Based on the figure's description in the main text and in the caption before that statement, I suggest labelling x axis to "time to reach threshold" and y axis to "eccentricity threshold". You clearly state in the methods that your experiments last for 10 Myr, so this labeling is quite confusing.

**Author's response:** In the experimental design we mention that all experiments 'span' the time period between 40 Ma and 30 Ma. We do not state that the experiments last for 10 Myr. Furthermore, we explain that the duration of the experiments where the influence of the orbital parameters is investigated is 2.4 Myr long (L160). Nevertheless, we agree with the reviewer to rename the axes towards 'Time to reach thresholds' and 'Eccentricity threshold' to increase the understanding of this figure and changed the figure accordingly. We also added arrows on the figure indicating the change in eccentricity threshold for changing $CO_2$ values to increase the figure clarity and added some explanations to the figure caption:

*The blue-green arrows indicate the decrease in eccentricity threshold for increasing $CO_2$ values to initiate glaciations and the orange arrows indicate the increase in the eccentricity threshold for decreasing $CO_2$ values to end glaciations.*

Figure 11: This is a very nice and illustrative figure - it gets the point across very easily. I just wonder what the red boxes around the dots mean?

**Author's response:** These boxes represent a 40 kyr interval (given by the width of the boxes, the height is meaningless). We have added the reference to the red boxes:

*Three different summer insolation values are indicated by the red dots that indicate the maximum insolation during a 40 kyr interval (indicated by the red boxes).*

Figures 13 and 14: "Simulations starting from bare bedrock" reads better, without the article. Note in Fig. 13 that you are referring to 2 simulations, so the caption should be in the plural as well.

**Author's response:** Corrected.

Figure 15: I really like this figure and think it is very illustrative. As in Fig. 6, it would be good to state in the caption what intervals the elevation contours represent.

**Author's response:** We added this information:

*Thin contour intervals are given every 250 m, while thick contour intervals are given each 1000 m.*

---

## Author Comment (AC2)

**Response to Anonymous Referee #2**

REVIEW OF 'HYSTERESIS AND ORBITAL PACING OF THE EARLY CENOZOIC ANTARCTIC ICE SHEET' BY VAN BREEDAM ET AL.

Van Breedam et al. quantify the CO2-thresholds for glaciation and deglaciation of the Early Cenozoic Antarctic ice sheet using an ice sheet model coupled to a climate model through a process emulator. They investigate the influence of the choice of bedrock topography, glacial isostatic adjustment, and orbital parameters (eccentricity and obliquity) on these thresholds. Their main findings are: 1) the choice of bedrock topography significantly affects both thresholds, but not the difference between them (hysteresis); 2) excluding glacial isostatic adjustment raises the CO2 level of glaciation but does not change the deglaciation threshold value; 3) the long eccentricity cycle has a significant impact on the timing of glaciation.

The topic of the study is interesting and timely, the methodology is sound, and the results are robust. My only concern is that the implications of the study are mostly left for the reader to deduce. In principle, the study achieves its stated aim of identifying the forcing needed to initiate and end a continental-scale glaciation, under various assumptions on bedrock topography, GIA and CO2. But what do the results actually mean for ephemeral glaciations prior to the EOT? What is the consequence of the hysteresis and orbital variations for the stability of the transiently evolving ice sheet? It could be worthwhile to include a (brief) discussion of proxy CO2 or ice volume data in this respect. The discussion section could be further improved for instance through a comparison, either present-day to Early Cenozoic Antarctica, or Antarctica to other ice sheets (the comparison to Greenland in the discussion section that is there, is a bit half-hearted). With some more discussion in this direction, the manuscript would in my opinion reach a wider audience.

**Author's response:** We thank the reviewer for the overall positive appraisal of the manuscript and for the suggestions to improve the significance. Also the discussion and the conclusions are rewritten to better grasp the implications of the research.

More specifically, we added a paragraph in the discussion on the implications of this study for ephemeral glaciations during the late Eocene. We also compared the early Cenozoic Antarctic ice sheet with a present-day forcing with the early Cenozoic Antarctic ice sheet, to assess the different climatic setting.

Furthermore, I have some recommendations to the authors for minor revisions to clarify the text and figures.

SPECIFIC COMMENTS PER SECTION:

§1 Introduction

L22-24:

The recent publication by Li et al. (2023) is also relevant here:

Li, Q., Marshall, J., Rye, C. D., Romanou, A., Rind, D., & Kelley, M. (2023). Global Climate Impacts of Greenland and Antarctic Meltwater: A Comparative Study. *Jounrla of Climate*, 1-40.

**Author's response:** We added the reference Li et al. (2023) that found a similar 50% decrease in AMOC, albeit for a different freshwater forcing.

L27, L32: 'favourable' and 'optimal'

The environmental conditions do not necessarily have to be favourable, and certainly not optimal, for glaciation to occur. They just have to be sufficient.

**Author's response:** Thanks for the remark. We changed 'favourable' to 'satisfactory' and 'optimal' to 'sufficient' in the revised manuscript.

L55-64:

Also worthy to mention that Pollard and DeConto (2005) did not use early Cenozoic Antarctic bedrock topography.

**Author's response:** We added this information and removed the information from the idealized orbital parameters, which is less crucial.

*L58-60:Pollard and Deconto (2005) determined the hysteresis of the early Antarctic ice sheet at the Eocene-Oligocene transition using an isostatically rebounded present-day bedrock topography and a matrix method where a limited number of climate model runs were performed based on end members in the forcing.*

L68:

Recent paleo-bedrock elevation reconstructions

**Author's response:** We made the change from 'bedrock elevation reconstruction' to 'paleo-bedrock elevation reconstruction'

§2 Model description and experimental set-up

§2.1:

Is ice-ocean interaction not important at all (i.e., no ice shelves)? How is the basal mass balance underneath the ice shelves calculated? And calving? Are sea level variations taken into account, if yes, how?

**Author's response:** The ice ocean interaction is taken into account, but is strongly simplified given the large unknowns about the ocean temperature and the translation of these temperatures and salinities into shelf melt rates. We opted for using a mean basal melt rate of 10 m per year. Calving physics are not explicitly included and the ice shelf breaks off when the ice thickness decreases below 120 m.

Which sliding law is used, and what about the grounding line physics? These are important factors, especially for deglaciation.

**Author's response:** We clarified the basal sliding and grounding line treatment as follows:

*The basal sliding velocity in AISMPALEO follows a Weertman relation and is proportional to the third power of the basal shear stress and inversely proportional to the height above buoyancy. The basal sliding coefficient is a constant multiplication factor for the basal sliding and equals 1.8 x 10⁻¹⁰ N⁻³yr⁻¹m⁸.*

*The basal sliding velocity in AISMPALEO follows a Weertman relation and is proportional to the third power of the basal shear stress and inversely proportional to the height above buoyancy. The basal sliding coefficient is a constant multiplication factor for the basal sliding and equals $1.8 \times 10^{-10} \ N^{-3}yr^{-1}m^{8}$.*

*The grounding line is a one grid cell wide transition zone between the grounded and floating ice where all the stress components contribute in the effective stress in the flow law.*

L90-91:

Influx of snow (from the atmosphere) and ice (from upstream), I think?

**Author's response:** Yes, we added this information.

L92:

So the input consists of daily temperatures and precipitation rates?

**Author's response:** Not completely, the PDD model uses the yearly sum of the mean daily temperatures above 0˚C to determine the melt potential, but monthly time steps are taken. Inputs are therefore monthly mean temperatures and precipitation rates. This information is added to the manuscript.

*In practice monthly time steps are sufficient to calculate the total amount of PDD's.*

L97-100:

Please mention the specifics of the GIA model.

**Author's response:** We add the following explanation for the GIA model, which is indeed an important component to describe in light of the isostasy sensitivity runs.

*The isostatic model consists of an elastic lithosphere with a flexural rigidity D of $10^{25}$ Nm (which is a measure of the strength of the lithosphere) on top of a viscous asthenosphere, to allow the crust to deform far beyond the local ice loading (Huybrechts, 2002). The vertical deflection of the lithosphere w is given by a fourth order differential equation (Eq. 2) Here, q is the ice load, $\rho_m$ is the mantle densitiy (3300 kg $m^{-3}$) and g the gravitational acceleration. This equation is solved using a Green's function. The viscous asthenosphere responds to the ice load with a relaxation time τ of 3000 years (Le Meur and Huybrechts, 1996).*

$$D\nabla^4 = q - \rho_m g \ w \quad (2)$$

L113-114:

The Wilson topographies are also used in the ISM, in higher resolution, I guess?

**Author's response:** That is indeed true. We explicitly mentioned it in the description and explained in more detail how we created the higher resolution topographies:

*L98-103:*

*The bedrock topography used in the climate model is the Wilson maximum bedrock topography and is representative for the Eocene-Oligocene transition (EOT) at 34 Ma (Figure 2). The minimum and maximum bedrock topographies are applied as a boundary condition in the ice sheet model at a 40 km resolution. In order to grasp the entire uncertainty, each ice sheet model grid cell takes on the lowest and highest value for respectively the minimum and maximum bedrock topography from the original higher resolution Wilson et al. (2012) dataset within each ice sheet grid cell.*

Figure 2:

It could be a valuable addition to show the difference between the two reconstructions.

**Author's response:** Thanks for the suggestion. We added a difference plot between the maximum and minimum bedrock topography reconstruction in Figure 2.

L121-122:

What vegetation is used for ice-free land, tundra (L500 seems to suggest this) or bare bedrock, and what is the albedo?

**Author's response:** The albedo varies between the 0.8 for ice/snow covered land and 0.2 for tundra. We added this as follows:

*L117-118:*

*The albedo varies between the discrete values of 0.8 for ice/snow and 0.2 for tundra.*

L127-128:

Worthy to mention Herrington and Poulsen (2011) here:

Herrington, A. R., & Poulsen, C. J. (2011). Terminating the Last Interglacial: The role of ice sheet–climate feedbacks in a GCM asynchronously coupled to an ice sheet model, Journal of Climate 25(6), 1871-1882.

**Author's response:** We added the reference Herrington and Poulsen (2011) as they found a similar coupling time step of 500 years, sufficient to capture climate-ice sheet feedbacks in a coupled ice sheet-climate model.

L137-139:

Winter precipitation may be lower, but judging from the temperatures summer precipitation will be almost all rain.

**Author's response:** We added this observation as follows:

*L140-141:*

*In winter, most precipitation falls as snow, while snowfall is limited to the highest elevated regions such as the Gamburtsev Mountains and Dronning Maud Land in summer.*

L147-149:

Mention that these CO2 bounds apply to this particular model set-up. You could add that these values are roughly in line with proxy data (see also my comment to L495-496).

**Author's response:** We added that these values are specific for our model set-up.

Figure 4:

Please make the black lines bracketing 34.2 and 31.8 Ma a little thicker, or preferably highlight this period in some other manner.

**Author's response:** We thank the reviewer for this suggestion. We added a grey shaded region to highlight the period between 34.2 and 31.8 Ma.

§3 Ice sheet hysteresis

L189-190:

It would be interesting to see what the difference in maximum ice thickness is between the two simulations (Wilson min and max), could you include a (supplemental) figure showing maps? And is the ice sheet in fact still land-terminating in this case (L198-199)?

**Author's response:** The ice sheets are mostly land terminating, expect for small regions along the West Antarctic ice sheet. The Wilson et al. (2012) bedrock topographies are significantly higher than a rebounded present-day bedrock topography because the bulk of the erosion has been taking place during the Miocene when the ice sheet is thought to have been very dynamic. A supplementary figure (Figure S1) showing the difference in ice sheet thickness between the simulations using the Wilson minimum and Wilson maximum dataset has been added.

Figure 6:

Why do you show this at 550 ppm CO2? I don't get that.

**Author's response:** Because the maximum ice sheet extent is reached at 550 ppmv.

Figure 7:

To aid comparison, it would be beneficial to have the same y-axis scales in both panels.

**Author's response:** Figure 7 has been adapted to have the same y-axis in both panels.

Figure 8:

Here as well, maps of maximum ice thickness (and maybe surface height) would be nice. I would think excluding GIA will lead to higher surface elevations, but only slightly because precipitation is depleted when the surface is raised. On the other hand, the ice will be less deep, because the bedrock topo remains higher.

**Author's response:** A supplementary figure (Figure S2) showing the difference in ice sheet thickness between the simulations including isostasy and excluding isostasy has been added.

L283-284:

I am not sure why the ice area is larger at the start of the no-GIA experiment.

**Author's response:** The ice area is larger shortly after the start of the experiments when a small ice sheet has been formed that has not depressed the bedrock, allowing for a larger accumulation area and hence a larger ice area. We rephrased this sentence:

*L324-326:*

*As it occurs, the initial temperature difference of about 0.5˚C between both experiments with $CO_2$ concentrations between 1000 and 1150 ppmv is due to the larger ice sheet area in the experiment that excludes isostasy.*

§4 Threshold dependency on orbital forcing

§4.1

This paragraph is not so clear to me. Which experiments do you discuss here? Judging from the text you keep the CO2 constant but vary all the orbital parameters. If that is the case, why are these experiments not described in §2?

**Author's response:** This is indeed what we do, the $CO_2$ is kept constant and the orbital parameters vary. These experiments are described in Table 2 of the experimental design (section 2.4).

L308: 'exceeding 0.03'

You mean below 0.03?

**Author's response:** We rephrased these sentences:

*L358-363:*

*The extreme eccentricity value of 0.064 that occurs after 2 Myr is not sufficient to melt the ice sheet once it has grown to a continental scale for a $CO_2$ forcing between 810 and 890 ppmv. This shows again the hysteresis effect of the Antarctic ice sheet: an eccentricity below 0.032 is enough to initiate continental-scale ice sheet growth for a $CO_2$ forcing up to 890 ppmv, while an eccentricity of >0.06 is not sufficient to make the ice sheet melt entirely after this. The extreme eccentricity of 0.064 influences the ice sheet volume for all simulations forced with a constant $CO_2$ concentration between 800 and 890 ppmv, but the peak insolation forced by the precessional cycle is too short-lived to melt the entire ice sheet (Figure S3).*

L298-299 (L317-318, L325)

Why has the ice sheet size changed after a few precessional cycles, is there a trend towards larger ice volume? Even at constant high CO2-levels?

**Author's response:** Indeed! The initial conditions (ice volume) have changed after a few precessional cycles, even for constant (high) $CO_2$ levels.

Figure 10:

The duration of the experiment is the time passed since 34.2 Ma, and it affects the background ice sheet size, right? The main text explains it well, but I must say the figure remains hard to interpret: I cannot see a clear pattern, particularly not for the glaciation threshold.

**Author's response:** I do agree that the figure is not that easy to read, but together with the text you understood it right. The overall pattern for the glaciation threshold is the following: the higher the $CO_2$ forcing, the longer it takes to achieve the threshold to full glaciation. So there is a clear pattern in the timing or the duration before the glaciation occurs, owing to the different initial state after a few precessional cycles. To initiate glaciation, a low eccentricity is beneficial because then the influence of the precession is attenuated. So, overall, you expect that the eccentricity threshold can be larger for lower $CO_2$ values and the eccentricity threshold should be lower for higher $CO_2$ values. We tried to increase the clarity of the figure by making these interpretations on the figure and added the following text to the figure caption:

*The blue and green arrows indicate the decrease in eccentricity threshold for increasing $CO_2$ values to initiate glaciations and the orange arrows indicate the increase in the eccentricity threshold for decreasing $CO_2$ values to end glaciations.*

L346-347:

The difference in timescale between glaciation and deglaciation mentioned later on (L375-376) is relevant here already, I believe.

**Author's response:** We agree with the reviewer and refer to Figure 12 already here.

Table 3:

You could add present-day values for comparison.

**Author's response:** We added the present-day austral summer insolation at 65˚S in Table 3 and the following text:

*L396-398:*

*For comparison, the present-day austral summer insolation (mean daily for DJF at 65˚S) is 439 Wm$^{-2}$. The eccentricity and the obliquity values resemble the cold orbital configuration, but the Earth is in perihelion during the austral summer today.*

L363-365, L366-367

Add 'not shown'.

**Author's response:** Done.

Figure 12:

And here again, showing maps (e.g., like Fig. 15 does) would be nice.

**Author's response:** We added maps showing the geometry right before the tipping point is reached at 900 ppmv for ice sheet growth (cold orbital configuration) and at 950 ppmv for ice sheet decline (warm orbital configuration) and added this information to the figure description.

L382-383:

So there is a double importance of ice area: the area itself (larger accumulation area), and the higher accumulation rate at the margins of the area. Moreover, the CO2 level affects the precipitation rates as well I think (warmer = more precip)?

**Author's response:** Not completely. Initially the accumulation is also larger inland in the Gamburtsev Mountains and decreases as the ice sheet develops. So there is no linear increase in the accumulation rate per unit area.

L387-389:

One could also think of an ice-volume-CO2 feedback loop: initial ice volume increase leads to lower CO2, which stimulates further glaciation.

**Author's response:** This feedback has been proposed to explain $CO_2$ variations during the Quaternary ice ages, possibly caused by changes in the ocean deepwater circulation or the Asian monsoon (Ruddiman, 2006). We do acknowledge that such a feedback mechanism could exist during the early stages of Antarctic glaciation, but the mechanisms behind it might be very different. Here we investigate the response of the ice sheet for prescribed $CO_2$ values.

Ruddiman, W. F.: Ice-driven $CO_2$ feedback on ice volume, Clim. Past, 2, 43-55, 2006.

Figure 14:

Could you add a color scale?

**Author's response:** The color scale is a measure of the ice thickness, but the ice thickness can also be read on the z-axis. Therefore, we believe it is not necessary to include a color scale to improve the figures quality.

§5 Discussion

L453-455:

Is colder ice flowing slower in fact a positive feedback? On the one hand, indeed, more ice will remain within the net accumulation zone. But on the other hand, expansion of the net accumulation zone due to surface uplift by inflowing ice is impeded.

**Author's response:** This is a good point since changes in the ice viscosity influences different processes and we did not attempt to disentangle the effect of ice temperature on the internal deformation and the ice sheet evolution. We decided to leave out this statement from the manuscript.

L460-470:

Hysteresis behaviour of the AIS is also quantified (albeit for the Miocene) in the appendix of Gasson et al. (2016):

Gasson, E., DeConto, R. M., Pollard, D., & Levy, R. H. (2016). Dynamic Antarctic ice sheet during the early to mid-Miocene. *PNAS 113(*13), 3459-3464.

**Author's response:** In this paragraph we discuss the effect of a certain change in ice sheet area on the glaciation and deglaciation. These ice-albedo effects have not really been quantified in the study from Gasson et al. (2016).

L471-480:

You could also compare to Abe-Ouchi et al. (2013), who found that instantaneous isostatic rebound obstructs deglaciation, in experiments of the Pleistocene Northern Hemisphere ice sheets.

Abe-Ouchi, A., Saito, F., Kawamura, K., Raymo, M. E., Okuno, J. I., Takahashi, K., & Blatter, H. (2013). Insolation-driven 100,000-year glacial cycles and hysteresis of ice-sheet volume. Nature 500(7461), 190-193.

**Author's response:** Isostatic rebound is not instantaneous but delayed, and as such Abe-Ouchi et al. (2013) states that during deglaciation, the isostatic rebound cannot prevent deglaciation. We also found that for deglaciation, the difference in ice volume response between the simulations including isostasy and excluding isostasy is negligible. We added the following text to the discussion:

*L545-548:*

*During deglaciation, our simulations have indicated no significant impact of isostasy on the threshold to deglaciation. Isostatic rebound has the potential to obstruct ice sheet melt, but the delayed response of the lithosphere to changes in ice loading is too slow to counteract the increasing snowline (Abe-Ouchi et al., 2013).*

§6 Conclusions

L495-496:

It should be noted that these values are very model-dependent, see Gasson et al. (2014). Maybe you could include a (brief) comparison to proxy data from around the EOT?

Gasson, E., Lunt, D. J., DeConto, R., Goldner, A., Heinemann, M., Huber, M., ... & Valdes, P. J. (2014). Uncertainties in the modelled CO2 threshold for Antarctic glaciation. Climate of the Past 10(2), 451-466.

**Author's response:** We thank the reviewer for the suggestion and added a brief comparison in the discussion.

*L508-511:*

*These $CO_2$ thresholds to initiate glaciations are 650 ppmv (minimum bedrock topography datset) and 870 ppmv (maximum bedrock topography dataset). Aside from the bedrock dataset dependence, the thresholds also depend on the climate model used. As shown in Gasson et al. (2014), the thresholds may vary between 560 and 920 ppmv when additionally climate model uncertainty is included.*

§ Code and data availability

Thank you for sharing the code of the emulator. I realize it is not required by the journal, but on a personal note I'd like to ask: Has the code of AISMPALEO also been made publicly available? If not, please do so. This facilitates transparency and reproducibility.

**Author's response:** The code of AISMPALEO is not publicly available, but people who are willing to use the code are welcome to contact us.

---

## Referee Report (RR1)

After reading through the author's responses and modified manuscript, I am happy to see that its quality and structure have improved immensely. The study is now much easier to follow, and the text flows much better. Presenting all experiments upfront, as well as a good description of all model setup components really help understanding what and how experiments were performed. In light of the new explanations and more detailed description of their methodology, I have further suggestions to improve the readability, and to openly discuss potential shortcomings of their model setup. This is a very interesting and thorough study, and it would be a shame if parts of it are not well appreciated by the reader just because some parts of the text are confusing, hence my effort in providing suggestions for rephrasing and more clarity in the explanations. Below I present these suggestions, using the line numbering in the **"track changes version"** of the manuscript.

**General**

One item that I would like to see in the discussion is a more open discussion (even if brief) about the fact that the uncertainty to model parameters is not explored (L178), and changes in sea level are not included (L214). It would be worth (and honest) to outline the setup shortcomings (within of course, the limitations of a poorly geologically constrained period and long simulated period), and how this could have affected the results.

**Line by line**

L116: The "anomalous heat convergence" is still not clear. The explanation provided in the authors in the response letter should be added to the manuscript. Again, keep in mind that CP has a broad audience, and the authors should not assume that all readers are familiar with how slab-ocean models work.

L117: sea ice, without hyphenation

L121: cite Wilson et al. (2012) here, at the first reference of the bedrock topography.

L122-125: This part is a bit confusing. The way it is written, it sounds like the model takes two different bedrock topographies. Is that correct, or are different experiments run using either the minimum or the maximum bedrock topography?

L135-136: From the use of "a unique combination", I understand that $\varepsilon$ and e are the same for all 100 ensemble members, and that the combination between the 20 different ice sheet geometries and different CO2 levels yield the 100 members. Based on the remainder of the paragraph, that does not seem to be the case. The paragraph reads odd when comparing L135-136 with L143-146 and Eq. 1, as it seems like your emulator takes all parameters into account (which is a good thing!). I understand that the details are provided in Van Breedam et al. (2021), but providing the number of values tested (as done for the ice sheet geometry) is some simple clarification that can be done in the sentences already written.

L160: representative of

L165-166: I suggest rephrasing for consistency and a better flow when reading: "while in summer snowfall is limited to the highest elevated regions, such as the Gamburtsev Mountains and Dronning Maud Land".

L176-178: From the description it seems like basal sliding is dependent on bed elevation, and/or ice thickness, but then there is also a spatially uniform coefficient? I strongly recommend clarifying this, perhaps as an equation similar to Eq. 1, highlighting what is dependent on (x,y) and what is not.

L179-180: I assume based on simulating present-day Antarctica with AISMPALEO? Good to clarify.

L182-183: Do you mean that you combine SIA and SSA, or that you apply the full Stokes equations over these grid cells? This needs to be clarified.

L188-189: I suggest the following rephrasing, for better readability: "Nevertheless, even for present-day simulations large uncertainties exist in how changes in ocean temperature and salinity affect melt rates below the ice shelves.". This statement, however, would benefit from a reference, e.g.: https://doi.org/10.5194/tc-16-4931-2022. Finally, the thickness threshold for calving mentioned as a response to Reviewer 2 should also be included in the model description.

L198-199: I believe this sentence belongs to the next paragraph.

L211: is it ice sheet initialisation or inception? Using "initiation" does not make this distinction clear

L217: remove "now", and replace "increasing" for "increased"

L227-229: I suggest rephrasing this sentence as follows, to make it clear that you are no longer referring to the runs shown in Table 1: "An additional set of runs (Table 2) explores the variation in orbital forcings to investigate the influence of insolation thresholds for ice sheet growth and decline in detail. In these runs, different values for the individual orbital parameters or the insolation are the control parameters explored (see Figure 1 and Table 2) and the $CO_2$ concentrations are kept constant".

L238: equilibrate faster -with- the forcing.

L320: are these temperature values absolute or relative? I assume the latter, in which case "by ~6℃ and ~8℃" would be more appropriate than "with ~6℃ and ~8℃".

L339: is it really 1080 ppmv? If I understood Fig. 8 correctly, it looks like it is ~1040 ppmv

L349: I would refrain from using "significantly" when not talking about statistical significance. This is something that is usually picked on by typesetters/copy editors, so best to change now.

L372-375: The authors provided in their response letter a valid justification for having both MAT_sur and MAT_clim. However, I still think the name MAT_clim is not intuitive for what it represents and is somehow misleading. I would suggest referring to it as MAT_corr or something similar, if the only difference between that and MAT_sur is the lapse-rate correction to bring it to a common reference level.

L378: If I understood this sentence correctly, I think it would be best phrased as "As it occurs, the initial temperature difference of about 0.5℃ between both experiments with CO2 concentrations of 1000 and 1150 ppmv is due to…"

L380: I am not sure what is meant by "the area of the continental scale ice sheet becomes nearly the same as the ice sheet extent". I struggle to see how the area of the ice sheet would not be the same as its extent? Is it just the fact that the ice sheet will ultimately occupy the entire extent of the continent?

L442 and elsewhere: using "melt" is not the most appropriate when referring to a complete meltdown of the ice sheet, as surface melt always happens regardless of whether your ice sheet vanishes or not. I would suggest using "demise" or "decline" instead, as used in other parts of the manuscript. Unless you indeed mean that there is no melt at all (i.e., SMB>0 always) below the thresholds discussed.

L461 and L465: "daily mean" as opposed to "mean daily".

L467: today -> at present

L459-461: Are these new experiments, or part of the ones listed in Table 2? If the latter, it is good to refer to the table here again.

L530: "at high eccentricity values" reads better.

L562: remove "ice" from "ice continent"

L570: snapshots

L573: If the authors agree that the message remains the same, this sentence could be rephrased as "Ice sheet hysteresis, as shown here and elsewhere (Oerlemans, 2002), is linked to the ice sheet geometry.". This would also clarify my confusion regarding sentence in the first review round.

L581: typo on "dataset"

L583: "When climate model uncertainty is also included". Using 'additionally' here reads odd.

L659: should the ice geometry and CO2 levels also be included as boundary conditions here?

Figure 11: use a shaded area as opposed to boxes (like in Fig. 4, which looks great) if the vertical axis is not supposed to be bounded. It should also make it easier to visualise the red

boxes over the red curves and red dashed lines. A small tip so that it does not get too loaded to the eyes is to apply a lighter shade of grey than done for Fig. 4, and also without the black lines around the box.

---

## Author Response (AR2)

**REVIEW OF 'HYSTERESIS AND ORBITAL PACING OF THE EARLY CENOZOIC ANTARCTIC ICE SHEET' BY VAN BREEDAM ET AL.**

The authors have sufficiently addressed my earlier concerns, and I am now ready to recommend publication of the manuscript (largely) as is. Especially the inclusion of more details on the ice-sheet model is appreciated.

Author's response: We thank the reviewer for the appreciation of our revised manuscript.

I just have some recommendations for small (technical) corrections:

Throughout the manuscript the reference to Huybrechts (1994b) should be Huybrechts (1994).

**Author's response: Corrected.**

Methods section:

- It is still not mentioned how calving is modeled.

- I believe the settings for geothermal heat flux and glacial isostatic adjustment are mostly representative for the East Antarctic ice sheet, which is by far the most important part in this study. For the West Antarctic ice sheet other values may be more appropriate though.

**Author's response:** There is no special treatment of calving fronts and iceberg calving occurs when the ice shelf front is thinner than 150 m. We added this information to the manuscript.

L167-168: There is no special treatment of calving fronts and iceberg calving occurs when the ice shelf front is thinner than 150 m.

L274: '~6 deg C and ~8 deg C' should be '-6 deg C and -8 deg C' I think.

Author's response: We state that the temperature needs to be lower by  $\sim 6^{\circ}$ C and  $\sim 8^{\circ}$ C, that indeed also corresponds to a temperature difference of  $-6^{\circ}$ C and  $-8^{\circ}$ C.

**L443-445:**

In the previous review I noted:

One could also think of an ice-volume-CO2 feedback loop: initial ice volume increase leads to lower CO2, which stimulates further glaciation.

Author's response: This feedback has been proposed to explain CO2 variations during the Quaternary ice ages, possibly caused by changes in the ocean deepwater circulation or the Asian monsoon (Ruddiman, 2006). We do acknowledge that such a feedback mechanism could exist during the early stages of Antarctic glaciation, but the mechanisms behind it might be very different. Here we investigate the response of the ice sheet for prescribed CO2 values.

Ruddiman, W. F.: Ice-driven CO2 feedback on ice volume, Clim. Past, 2, 43-55, 2006

New comment:

I understand your modeling strategy, but I still think the potential for the existence of such a feedback in reality could be mentioned in the discussion as it could influence the time scale for glaciation and deglaciation as well.

Author's response: We briefly mention the existence of the feedback now in the discussion.

L522-524: One of the possible causes of the  $CO_2$  variations during the early Cenozoic could be the build-up of the ice sheet itself, which causes a positive feedback loop with increased ice volume leading to lower  $CO_2$  levels and amplified glaciations (Ruddiman, 2006).

**Response to reviewer #2**

After reading through the author's responses and modified manuscript, I am happy to see that its quality and structure have improved immensely. The study is now much easier to follow, and the text flows much better. Presenting all experiments upfront, as well as a good description of all model setup components really help understanding what and how experiments were performed. In light of the new explanations and more detailed description of their methodology, I have further suggestions to improve the readability, and to openly discuss potential shortcomings of their model setup. This is a very interesting and thorough study, and it would be a shame if parts of it are not well appreciated by the reader just because some parts of the text are confusing, hence my effort in providing suggestions for rephrasing and more clarity in the explanations. Below I present these suggestions, using the line numbering in the **"track changes version"** of the manuscript.

**Author's response:** We thank the reviewer for the positive appraisal of our revised manuscript and for the suggestions to improve the readability.

**General**

One item that I would like to see in the discussion is a more open discussion (even if brief) about the fact that the uncertainty to model parameters is not explored (L178), and changes in sea level are not included (L214). It would be worth (and honest) to outline the setup shortcomings (within of course, the limitations of a poorly geologically constrained period and long simulated period), and how this could have affected the results.

Author's response: We have added the following paragraph related to the ice sheet model uncertainty to the discussion:

L585-593: Despite the performance of a substantial amount of sensitivity experiments related to the orbital and  $CO_2$  forcing, we did not perform a sensitivity analysis of the ice sheet parameter uncertainty. Ice sheet model parameters such as the enhancement factor, the basal sliding coefficient or the flow rate factor each have their uncertainty and they are tuned to reproduce the present-day Antarctic ice sheet. Also the parameters related to the isostatic model are not explored. For instance, the relaxation time of the asthenosphere and the flexural rigidity of the lithosphere add another degree of freedom. The most important parameters influencing ice sheetclimate model feedbacks are however parameters related to the albedo (Gandy et al., 2023). Because our modelling approach on very long timescales significantly improved the representation of albedo changes resulting from changes in the ice sheet extent (Van Breedam et al., 2021a), we believe that the ice sheet parameter uncertainty is of secondary importance here. We also added the following justification to the experimental design:

L191-193: Sea-level changes (sea-level fall when the ice sheet is growing) are not included as an additional forcing because the number of grid points that become land-based when sea-level would fall by 70 m are very limited.

**Line by line**

L116: The "anomalous heat convergence" is still not clear. The explanation provided in the authors in the response letter should be added to the manuscript. Again, keep in mind that CP has a broad audience, and the authors should not assume that all readers are familiar with how slab-ocean models work.

Author's response: We added the following information:

L93-95: The slab ocean model equilibrates with the atmosphere in a 50 m thick layer where heat is exchanged with the atmosphere. In slab ocean-atmosphere models, an anomalous heat convergence is added to the ocean to mimic the real influence of oceanic circulation below the slab layer.

L117: sea ice, without hyphenation

Author's response: Corrected.

L121: cite Wilson et al. (2012) here, at the first reference of the bedrock topography.

**Author's response: Done.**

L122-125: This part is a bit confusing. The way it is written, it sounds like the model takes two different bedrock topographies. Is that correct, or are different experiments run using either the minimum or the maximum bedrock topography?

**Author's response:** Different ice-sheet experiments are run using either the minimum or the maximum bedrock topography, but the minimum and maximum bedrock topography are regridded to the lower-resolution ice sheet model grid as explained. For each of the two Wilson topographies the maximum and minimum bedrock height within the lower-resolution ice sheet grid cell is taken to quantify the bedrock uncertainty.

L135-136: From the use of "a unique combination", I understand that  $\varepsilon$  and e are the same for all 100 ensemble members, and that the combination between the 20 different ice sheet geometries and different CO2 levels yield the 100 members. Based on the remainder of the paragraph, that does not seem to be the case. The paragraph reads odd when comparing L135-136 with L143-146 and Eq. 1, as it seems like your emulator takes all parameters into account (which is a good thing!). I understand that the details are provided in Van Breedam et al. (2021), but providing the number of values tested (as done for the ice sheet geometry) is some simple clarification that can be done in the sentences already written.

Author's response: No, a unique combination means that each parameter combination is unique, so different.

L160: representative of

**Author's response: Done.**

L165-166: I suggest rephrasing for consistency and a better flow when reading: "while in summer snowfall is limited to the highest elevated regions, such as the Gamburtsev Mountains and Dronning Maud Land".

Author's response: Adapted.

L176-178: From the description it seems like basal sliding is dependent on bed elevation, and/or ice thickness, but then there is also a spatially uniform coefficient? I strongly recommend clarifying this, perhaps as an equation similar to Eq. 1, highlighting what is dependent on (x,y) and what is not.

**Author's response:** That is true, basal sliding is spatially dependent on bed elevation and ice thickness, but calculated for a spatially uniform basal sliding coefficient. We do not see any added value of adding the equation here.

L179-180: I assume based on simulating present-day Antarctica with AISMPALEO? Good to clarify.

Author's response: That information is added.

L182-183: Do you mean that you combine SIA and SSA, or that you apply the full Stokes equations over these grid cells? This needs to be clarified.

Author's response: No, we do not combine SIA and SSA, neither the full Stokes equations, at the grounding line all stress components contribute to the effective stress as we already mentioned.

L188-189: I suggest the following rephrasing, for better readability: "Nevertheless, even for present-day simulations large uncertainties exist in how changes in ocean temperature and salinity affect melt rates below the ice shelves.". This statement, however, would benefit from a reference, e.g.: https://doi.org/10.5194/tc-16-4931-2022. Finally, the thickness threshold for calving mentioned as a response to Reviewer 2 should also be included in the model description.

Author's response: We thank the reviewer for the suggestion, we rephrased the sentence accordingly and added the reference Burgard et al. (2022).

L198-199: I believe this sentence belongs to the next paragraph.

Author's response: The sentence has been moved.

L211: is it ice sheet initialisation or inception? Using "initiation" does not make this distinction clear

Author's response: We mean inception and replaced the word 'initiation' for 'inception'.

L217: remove "now", and replace "increasing" for "increased"

Author's response: This has been changed.

L227-229: I suggest rephrasing this sentence as follows, to make it clear that you are no longer referring to the runs shown in Table 1: "An additional set of runs (Table 2) explores the variation in orbital forcings to investigate the influence of insolation thresholds for ice sheet growth and decline in detail. In these runs, different values for the individual orbital parameters or the insolation are the control parameters explored (see Figure 1 and Table 2) and the CO2 concentrations are kept constant".

Author's response: Thank you for the suggestion. We rephrased the sentence.

L238: equilibrate faster -with- the forcing.

Author's response: Corrected.

L320: are these temperature values absolute or relative? I assume the latter, in which case "by  $\sim$ 6°C and  $\sim$ 8°C" would be more appropriate than "with  $\sim$ 6°C and  $\sim$ 8°C".

**Author's response: Corrected.**

L339: is it really 1080 ppmv? If I understood Fig. 8 correctly, it looks like it is ~1040 ppmv.

**Author's response: Corrected.**

L349: I would refrain from using "significantly" when not talking about statistical significance. This is something that is usually picked on by typesetters/copy editors, so best to change now.

**Author's response: We removed the word 'significantly'.**

L372-375: The authors provided in their response letter a valid justification for having both MAT\_sur and MAT\_clim. However, I still think the name MAT\_clim is not intuitive for what it represents and is somehow misleading. I would suggest referring to it as MAT\_corr or something similar, if the only difference between that and MAT\_sur is the lapse-rate correction to bring it to a common reference level.

**Author's response:** We still believe the MAT\_clim is representative for what it actually is representing: the climatological mean annual temperature that is not taking into account the temperature changes resulting from changes in the height of the ice sheet.

L378: If I understood this sentence correctly, I think it would be best phrased as "As it occurs, the initial temperature difference of about 0.5°C between both experiments with CO2 concentrations of 1000 and 1150 ppmv is due to…"

Author's response: No, it is not only for  $CO_2$  concentrations of 1000 and 1150 ppmv, but for all  $CO_2$  concentrations in between as well.

L380: I am not sure what is meant by "the area of the continental scale ice sheet becomes nearly the same as the ice sheet extent". I struggle to see how the area of the ice sheet would not be the same as its extent? Is it just the fact that the ice sheet will ultimately occupy the entire extent of the continent?

Author's response: The sentence is reformulated as follows:

L328-330: After the transition to a continental scale ice sheet for a  $CO_2$  concentration below 850 ppmv, this difference between both simulations is negligible because the area of the continental scale ice sheet is ultimately bounded by the size of the Antarctic continent.

L442 and elsewhere: using "melt" is not the most appropriate when referring to a complete meltdown of the ice sheet, as surface melt always happens regardless of whether your ice sheet vanishes or not. I would suggest using "demise" or "decline" instead, as used in other parts of the manuscript. Unless you indeed mean that there is no melt at all (i.e., SMB>0 always) below the thresholds discussed.

Author's response: The word 'melt' is replaced by 'decline', 'demise, 'disintegrate' and 'make the Antarctic ice sheet disappear'.

L461 and L465: "daily mean" as opposed to "mean daily".

Author's response: Corrected.

L467: today -> at present

Author's response: Corrected.

L459-461: Are these new experiments, or part of the ones listed in Table 2? If the latter, it is good to refer to the table here again.

**Author's response:** These specific experiments are not listed in Table 2. We added new labels in Table 2 (a-f) and refer to the table when the specific experiment is performed.

L530: "at high eccentricity values" reads better.

Author's response: Corrected.

L562: remove "ice" from "ice continent"

Author's response: Done.

L570: snapshots

Author's response: Done.

L573: If the authors agree that the message remains the same, this sentence could be rephrased as "Ice sheet hysteresis, as shown here and elsewhere (Oerlemans, 2002), is linked to the ice sheet geometry.". This would also clarify my confusion regarding sentence in the first review round.

Author's response: We thank the reviewer for the suggestion and adapted the sentence accordingly.

L581: typo on "dataset"

Author's response: Corrected.

L583: "When climate model uncertainty is also included". Using 'additionally' here reads odd.

**Author's response: Corrected.**

L659: should the ice geometry and CO2 levels also be included as boundary conditions here?

**Author's response:** We do not define them here as boundary conditions but as forcing/feedback, while the bedrock topography is regarded as a boundary condition.

Figure 11: use a shaded area as opposed to boxes (like in Fig. 4, which looks great) if the vertical axis is not supposed to be bounded. It should also make it easier to visualise the red boxes over the red curves and red dashed lines. A small tip so that it does not get too loaded to the eyes is to apply a lighter shade of grey than done for Fig. 4, and also without the black lines around the box.

Author's response: We thank the reviewer for this very good suggestion and adapted the figure accordingly.